# A quantum-mechanical framework for million-atom scale biological systems
Luc Wieners & Martin E. Garcia ✉

Quantum-mechanical simulations provide the most fundamental description of matter, yet their computational cost commonly limits applications to systems containing at most thousands of atoms. Here, we present an all-electron quantum-mechanical framework focused on very fast calculations up to the multimillion-atom regime which is achieved by scaling down the accuracy of the framework while still maintaining agreement with experimental results. By combining algorithmically optimised Hartree-Fock with divide-and-conquer, using a minimal basis set and truncating long-range interactions, our approach efficiently handles million-atom structures while remaining accessible for smaller computation clusters and saving energy due to fast run times. We demonstrate this approach on very large biological systems, including a bacteriophage in water, totalling over 150 million electrons, representing, to our knowledge, the largest Hartree-Fock calculation performed to date. Our framework allows computing spectral data for DNA and drugs and enables protein structure assessments in strong agreement with structure evaluations by AlphaFold.

The behaviour of proteins and other biomolecules is mainly governed by the quantum-mechanical character of their electrons. Accurately capturing the resulting interactions is essential for predicting molecular properties, obtaining spectroscopic data and advancing drug design. However, the extreme computational cost of quantum calculations has often limited their application to smaller systems. Here, we modify an all-electron quantum-mechanical method, enabling electronic structure calculations on biological systems up to millions of atoms at a reduced accuracy to drastically lower computational costs.

As a proof of concept, we apply this approach to entire proteins and large biomolecular assemblies, including a complete bacteriophage[1] in a solution containing over 45 million atoms. Additionally, we show that atomic energies computed for AlphaFold-predicted protein structures strongly correlate with AlphaFold's confidence scores[2], providing a new quantum-based validation metric. The method's efficiency also allows the accurate prediction of spectroscopic properties for biomolecules previously out of reach for first-principles techniques. We present computed spectra for DNA[3,4] and the anticancer drug Actinomycin[5,6], involving hundreds to thousands of atoms, in close agreement with experimental measurements.

## Results

### Large-scale quantum mechanics

Biomolecules, including proteins, nucleic acids and large assemblies, underpin nearly all processes of life. The rich range of their structures and functions arises from their ability to undergo conformational transitions and biochemical reactions, the latter often involving the formation or breaking of covalent bonds and the redistribution of electrons. While classical molecular mechanics methods[7–9] can successfully describe large-scale conformational changes, a fully accurate treatment of processes that require bond rearrangements, such as enzyme catalysis, photoexcitation, or protein production, must account for the quantum nature of electrons. To tackle these problems, hybrid quantum mechanics/molecular mechanics (QM/MM) methods[10,11] have become a popular compromise, applying a quantum description to a small reactive site while retaining classical models elsewhere. However, this approach cannot fully capture the rich electronic effects that emerge across large biomolecular assemblies. A fully quantum-mechanical treatment at this scale has remained challenging due to the extreme computational cost associated with large-scale calculations.

Here we use a framework based on the all-electron Hartree–Fock method that makes it possible to perform first-principles electronic structure calculations[12,13] on whole proteins and large biomolecular[14–22] complexes at manageable computational costs. First, we identify and discard those electron repulsion integrals that have a small influence on the final results, thereby reducing computational effort. Second, we employ a divide-and-conquer strategy to partition large systems into manageable segments, allowing us to treat their electronic structure quantum-mechanically (see Supplementary Fig. 6).

This framework opens up a path toward cost-efficient, fully quantum-mechanical simulations of complex biomolecular systems, the possibility of shedding light on their mechanisms, properties, and functions and paves the

Institute of Physics, University of Kassel, Kassel, Germany. ✉e-mail: m.garcia@uni-kassel.de

way for future applications in structural biology, biochemistry, drug design and interaction of external electric, magnetic and electromagnetic fields with proteins and DNA.

## Strengths and limitations of the method

The presented framework is focused on increasing the computational speed as much as possible while still maintaining results in agreement with other data. This can only be done at the cost of accuracy but leads to speed increases of orders of magnitude. Consequently, the focus of the method lies on calculations for systems which would remain inaccessible with methods of higher accuracy and studies where no high-accuracy results are necessary or situations with reduced computational resources available. In the following we describe the used methods and approximations which are necessary to achieve the increase in computation speed.

First, we use the minimal basis at STO-3G accuracy to decrease the number of basis functions and Gaussian functions as much as possible. The minimal basis is the natural choice for our approach focused on the computational speed—compared to larger split-valence and polarisation basis sets, our approach reduces the number of electron repulsion integrals by orders of magnitude, significantly lowering memory requirements and diagonalization time. Relative to state-of-the-art high-accuracy Hartree–Fock implementations[15], we achieve a computational speedup of more than three orders of magnitude in floating-point operations per atom, comparing R-Peak values of the respective computing clusters.

Due to the use of the Hartree-Fock method, electronic correlation is inherently neglected. Hartree-Fock calculations, especially with a minimal basis, profit largely from the sparsity of the atoms in biological systems as their density is lower than for solids which is even further amplified if water is included in the system. The described low atom density and approximately one half (or one third for water) of the atoms being hydrogen atoms makes the calculation of the electron repulsion integrals very fast as their number is largely reduced under these conditions and compared to DFT avoids the use of large integration grids since no numerical integration is involved.

Finally, we discard low-importance density matrix elements and long-range interactions by applying a short Coulomb cut-off for electron repulsion integrals and nucleus-nucleus interactions. In the divide-and-conquer approach the latter is forced due to its realisation (see section below). If no divide-and-conquer is used, as for the spectral calculations, the Coulomb cut-off can be scaled to higher values to account for long-range interactions. In this work, we work with a low Coulomb cut-off of 1 nm regardless as a higher cut-off showed no significant influence in the applications that were investigated (details in 'Methods' section).

It should be mentioned that the use of the divide-and-conquer approach is focused on the applications in this work, for other areas—especially where higher accuracy is needed—the truncation of long-range interactions could potentially compromise the results. For systems with hundreds to a few thousand atoms, where no divide-and-conquer is needed, the low Coulomb cut-off leads to similar problems. In cases where smaller systems are studied at a higher accuracy, it is therefore necessary to work without divide-and-conquer and no (or a larger) Coulomb cut-off. Test results can be found in the section 'Further testing with molecular dynamics' in Methods.

The presented results in the following sections show that calculations at a reduced accuracy can give results in good agreement with experiments. This opens up the pathway to investigating the methodology in more detail as its clear strength is the aforementioned strong reduction of computation time. Using as little computational resources as possible is favourable to save energy, especially considering the extreme energy demands of supercomputing clusters. Furthermore, an approach with reduced accuracy is helpful in circumstances where no high-accuracy results are needed or to study large systems that remain inaccessible with high-accuracy methods.

Additionally, the presented framework in this work can also be run with small computational resources. While the mentioned calculation of a bacteriophage was run at a supercomputing cluster, our approach would also allow computations up to a million atoms on a single compute node within a few days.

Finally, for the case of spectral calculations (see corresponding section) the fast-running RTHF algorithm allows fully quantum-mechanical calculations of molecules at previously not studied sizes. The needed computational resources are very low and for systems up to a few hundred atoms, these calculations can even be run on personal computers or laptops.

## Quantum divide and conquer in biomolecules

To enable all-electron quantum-mechanical calculations on large biomolecular systems, we use a divide-and-conquer strategy[16,20,23,24] that avoids an unfavourable scaling. Instead of solving the Hartree–Fock equations for the whole molecule at once, which typically scale cubically with system size, we split the system into overlapping clusters of manageable size. Each of these clusters is treated separately in a Hartree–Fock framework, yielding an order-N scaling of computational effort[12,14,17,18,23–32] and allowing the procedure to be easily parallelised across multiple processors. Every cluster comprises a central 'core' region containing the atoms of primary interest, while a surrounding 'buffer' region captures the effects of its environment. Only the electron density of the core is kept for assembling the total density, while the buffer guarantees that boundary effects and medium-range interactions are adequately represented. The clusters are then merged by interpolating their density matrices, yielding a total electron density for the entire system (see 'Methods').

To generate the subsystems used in the divide-and-conquer scheme, we apply different strategies, depending on the structural characteristics of the biomolecule. For heterogeneous systems with large empty regions we employ the k-means clustering algorithm to group nearby atoms into spatially localised clusters that serve as the core regions for quantum calculations (see Fig. 1b) for an exemplary (protein complex[33], visualised using ChimeraX[34]). On the other hand, for more homogeneous systems (as for example protein complexes in a solution (Fig. 1c)), we use a three-dimensional grid to define the partitioning.

Once the core regions are identified, we expand each cluster by adding surroundings of atoms within a fixed cut-off distance (typically of 8–10 Å), to form the buffer region which captures the long-range effects. However, adding atoms just based on their distance often leads to an artificial cutting of covalent bonds, introducing unphysical boundary conditions. To overcome this problem, we apply chemically informed rules: if a double bond is intersected, both bonded atoms are included in the cluster. For a single bond, the cut atom is replaced by a hydrogen atom positioned at the typical length of the bond X-H for a corresponding atom of species X. Bond character is determined from atomic distances, ensuring that all cluster boundaries preserve chemically realistic connectivity.

Achieving linear scaling is essential for large-scale quantum-mechanical calculations. But to make such calculations practical, it is also important to reduce the prefactor of the now linear complexity curve as much as possible. This requires not only efficient partitioning but also highly optimised algorithms for performing the Hartree–Fock all-electron calculations within each subsystem. We therefore use a modified screening algorithm[25,26,35] which estimates the relevance of the interaction of two wave functions (which together form an electronic density) based on their overlap. In addition to this, we use a cut-off for the Coulomb interaction of two densities (for example 10 Å). In combination these methods drastically reduce the number of evaluated two-electron repulsion integrals (ERIs) which usually are the computational bottleneck of Hartree–Fock calculations, since they scale with the fourth power of the system size.

To illustrate the efficiency gains, we applied it to the retinal protein Rhodopsin (PDB 1H68)[36], which consists of 3439 atoms. In the STO-3G basis set, the theoretical number of ERIs is approximately $1.06 \times 10^{16}$, of which $1.33 \times 10^{15}$ have distinct values. By applying a filtering scheme based on relevant densities (see 'Methods'), we reduced the number of significant ERIs to $1.57 \times 10^{11}$. A second filtering step, specific to the Coulomb interaction, further reduced this value to just $1.33 \times 10^{9}$ ERIs, a decrease of six orders of magnitude with respect to the full set. For the efficient computation

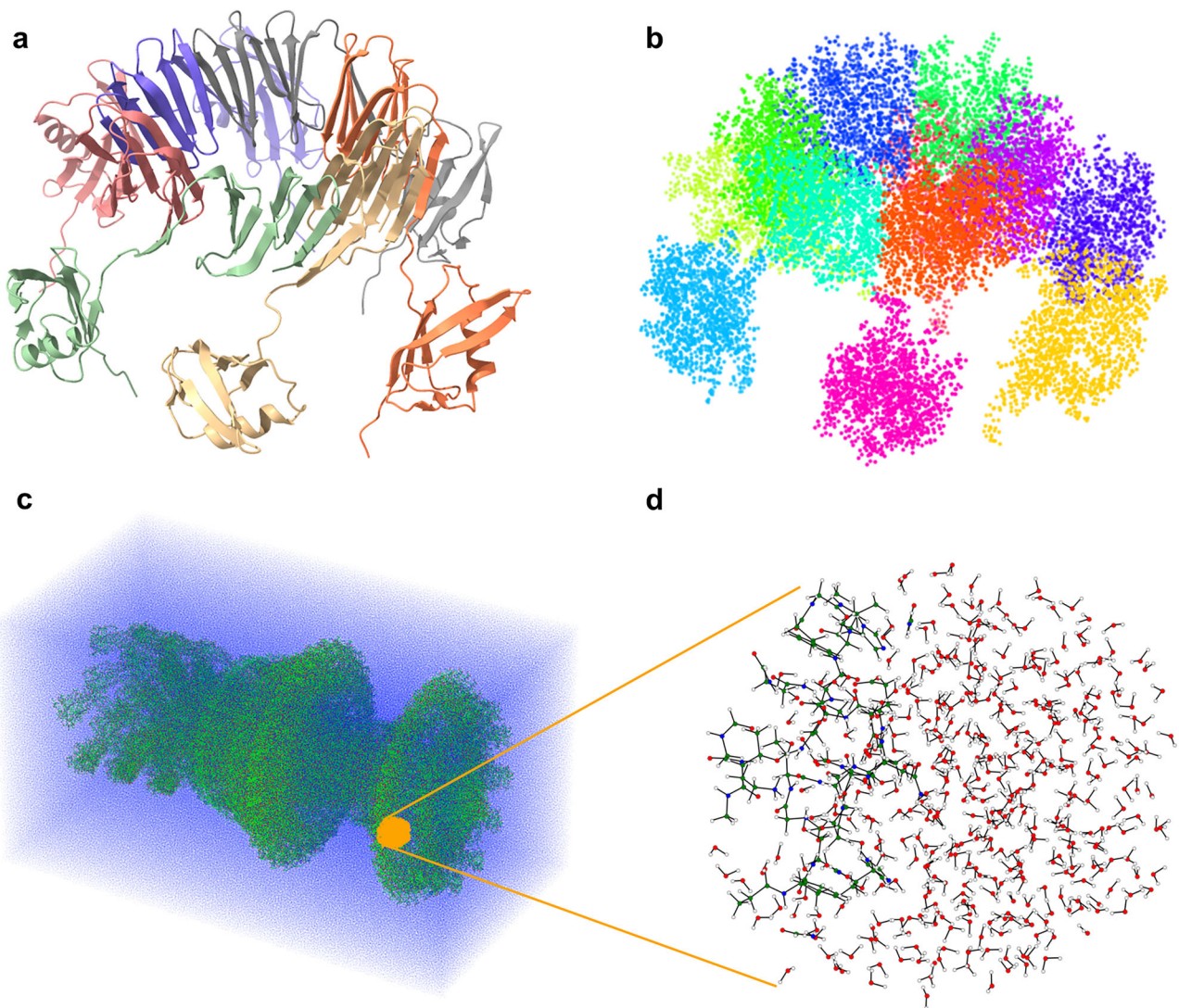

**Fig. 1 | Divide-and-conquer clustering for linear-scaling quantum calculations.** **a** The protein complex (PDB 2M3X)[33] is shown as an example system. **b** The molecule is spatially partitioned into 12 clusters using the k-means algorithm. **c** For larger systems in a solution, such as a flagellar motor in water, the individual clusters

are obtained by subdividing the system using a three-dimensional grid. **d** Corresponding grid-based clusters are shown. Visualisation of the protein structure in **a** was done with ChimeraX[34].

of the remaining ERIs, we used a Gaussian lobe function expansion[37], which enables highly parallelizable algorithms and contributes to the overall scalability of the method.

In addition to the computation time, memory requirements are a critical factor in Hartree–Fock calculations. The need to store the ERIs usually presents the main memory bottleneck, as even a single system may easily involve more than several billion ERIs, exceeding the storage capacity of typical high-performance computing nodes. The used filtering strategy overcomes this limitation by discarding integrals that negligibly contribute to the final result, dramatically reducing memory usage while preserving accuracy.

When applying a divide-and-conquer strategy to systems with more than 100,000 atoms[14–16,24,27,32], the number of subsystems can itself become a limiting factor, particularly in terms of memory requirements on high-performance computing facilities. Because subsystems interact, their data must remain accessible to one another, typically requiring all clusters to be evaluated in parallel, which can overwhelm available resources. We overcome this problem by adding overlaps between neighbouring subsystems, allowing them to be processed sequentially rather than simultaneously. This approach removes a major scaling bottleneck (see 'Methods') and enables

quantum-mechanical calculations on systems with over 10 million atoms at comparably low computational costs and with much fewer computation nodes used due to the sequential processing.

## Electronic structure of a bacteriophage

Using our modified Hartree–Fock formalism, we performed quantum-mechanical calculations on entire biophysical systems comprising millions of atoms, including one of the largest systems treated at this level of theory. As benchmarks, we selected three structurally distinct complexes in explicit water: a bacterial flagellar motor (PDB 8WL2)[38], a vault ribonucleoprotein particle (PDB 7PKR, see Supplementary Fig. 7)[39] and a complete Staphylococcus aureus bacteriophage (P68, PDB 6Q3G)[1]. These systems consist of approximately 4.37, 13.5 and 45.1 million atoms, respectively. This corresponds to 14.9, 45.4 and 151.3 million electrons, respectively, including those of the water molecules.

The bacteriophage P68 comprises 668 protein subunits, together containing 133,777 amino acids (2,148,795 atoms). The structure was solvated in a water box extending 10 nm beyond the molecular surface in all directions, yielding a total system size of $67.2 \times 78.8 \times 86.8$ nm³. To enable the calculation, we divided the system into 241,920 overlapping subsystems,

each with a central region of $(12.5\,\text{Å})^3$ and 8 Å overlap. Hartree–Fock calculations were carried out independently for each subsystem, and the global electronic density was reconstructed by interpolating the nonzero elements of the subsystem density matrices. Orbital energies were also retained for subsequent analysis. The full calculation was performed on 150 compute nodes (Intel Xeon Platinum 9242, 96 cores/node) and completed in 12 h, amounting to 173,000 CPU core hours which translate to 32 quintillion $(3.2 \times 10^{19})$ R-peak floating-point operations. A uniform grid with 1.5 bohr spacing was used to evaluate the total electron density, shown in Fig. 2 and Fig. 3.

## Quantum optical spectra of biomolecules

A defining strength of quantum-mechanical approaches is their ability to predict optical properties that are inaccessible to classical models. Absorption spectra, for example, arise from electronic excitations and therefore require a fully quantum description of the electronic structure. Using our Hartree–Fock framework in a real-time time-dependent Hartree–Fock (RT-TDHF) implementation[40], as the RTHF approach offers more favourable scaling for large systems than linear-response methods, we computed UV/Vis absorption spectra[13] of large biomolecules[41]. The described integral filtering and use of the STO-3G basis allows the calculation of the optical response of systems comprising several hundred to a few thousand atoms, a regime typically out of reach for direct quantum spectral calculations.

We applied this method to obtain the absorption spectra of DNA[3,4] and the anticancer drug Actinomycin D[5] as well as a DNA-Actinomycin D complex[6] and compared to experimentally obtained spectral data[42–44] (see Fig. 4). These systems are—to our knowledge—the largest systems for which theoretical UV/Vis spectral data has been obtained from first principles.

## Quantum energies and confidence scores

AlphaFold[2] has revolutionised protein structure prediction, offering remarkably accurate models accompanied by residue-level confidence scores known as pLDDT (predicted local distance difference test) value[45]. These scores range from 0 (high uncertainty) to 100 (high confidence), offering an estimate of how closely a predicted structure matches the true, native conformation.

Using the described linear-scaling Hartree–Fock approach, we calculate atomic energies of entire protein structures efficiently and with quantum-mechanical accuracy. Because proteins tend to fold into structures that minimise their free energy[46], local atomic energies provide a direct, physics-based measure of structural stability. Regions with low atomic energies are expected to be close to their native, folded state, whereas high-energy regions are a sign of instability or disorder. This allows us to obtain a measure for the assessment of protein structures based on first-principles methods (see Supplementary Fig. 8).

To evaluate this principle, we studied three AlphaFold-predicted proteins: Evasin P1126 (PDB AF_AFA0A023FF81F1), MHC Class II Beta Chain (PDB AF_AFA0A023IKK2F1) and Friend of Echinoid Isoform H (PDB AF_AFA0A023GPK8F1), obtained from the RCSB PDB[47] and containing 90, 287 and 1,447 amino acids, respectively. Each structure was solvated with a 10 Å shell of water to create a more realistic environment and to avoid artificial edge effects. The resulting system sizes ranged from ~9000 to over 115,000 atoms, with corresponding runtimes from under an hour to 14 h on a single compute node. Atomic energies were calculated and normalised by element type (using only valence basis functions) and further smoothened using a suitable filter to remove short-range noise (see 'Methods'). These energies were rescaled and clipped to match the [0, 100] range of pLDDT scores, enabling a direct comparison. Strikingly, we found a strong correlation between quantum-mechanical atomic energies and AlphaFold's pLDDT scores across all three proteins (Fig. 5) with Pearson correlation coefficients of 0.953, 0.914 and 0.946, respectively. Regions with low energies aligned closely with AlphaFold's high-confidence predictions, while higher energies corresponded to lower pLDDT values. A similar trend was observed when using simplified energy models based on ionic

contributions alone, although with reduced accuracy. In comparison to classical heuristics such as counting neighbouring atoms to evaluate protein structures via atom densities, the quantum-mechanical approach exhibited a markedly stronger correlation with pLDDT scores, stressing the value of using first-principles methods in assessing protein folding predictions (see Supplementary Fig. 9).

Our results suggest that AlphaFold's predictions not only approximate correct protein structures, but also implicitly capture aspects of the quantum-mechanical energy landscape—despite not being trained on quantum data. This opens the door to using first-principles calculations to validate and interpret machine-learning-based structure predictions and provides an independent, physics-grounded confidence metric.

## Discussion

The approach presented here enables advances across biology and medicine by making quantum-mechanical descriptions of entire biological assemblies accessible at a reduced accuracy, but at equally reduced computational costs. Since the results for the two presented applications (UV/Vis absorption spectra and predicted protein structure evaluation) show good agreement with other data regardless of the reduction in accuracy, we propose more investigations of QMs with downscaled accuracy. Next to simplifying the study of large-scale systems, this approach could potentially be more sustainable due to less energy consumption and more accessible due to the possibility of studying larger systems on small computating clusters, even on single compute nodes.

As a proof of concept, we computed the charge density of a complete bacteriophage and the absorption spectrum of a large DNA fragment. The efficiency of the method is quantified by computation-time tracking: for the bacteriophage calculation, approximately 32 quintillion R-peak floating-point operations are required, which can be achieved in less than a minute by today's leading supercomputers. Furthermore, spectral calculations for DNA and drug molecules show very good agreement with experimental data, confirming that the presented framework can accurately predict complex properties of large systems beyond the reach of conventional approaches.

The calculation of atomic energies via the presented modified Hartree–Fock scheme offers a physically grounded metric for assessing protein structure predictions, independent of machine learning biases. Given its high computational efficiency, this approach provides a complementary method to evaluate predicted models—especially in regions with limited training data or in intrinsically disordered proteins[48]. The ability to localise energetically unfavourable regions could support the identification of misfolded domains or regions of structural uncertainty.

Beyond structure assessment, the fast-running RT-TDHF algorithm enables the calculation of optical absorption spectra for biomolecular systems at a size previously not studied with ab initio methods. This includes drug molecules and their interactions with DNA or proteins, as demonstrated for Actinomycin D. For smaller molecules, the low computational requirements of the modified Hartree–Fock algorithm even permit UV/Vis spectra to be calculated on personal computers, potentially broadening access to theoretical spectroscopy.

In addition, the direct access to the electronic density enables further applications[49]. Dipole moments, for instance, can be determined with higher accuracy than via classical approximations[50]. More importantly, the computed electronic densities allow direct comparison with x-ray diffraction experiments, making it possible to theoretically reconstruct electron density maps at quantum-mechanical precision (see Supplementary Fig. 10). This is particularly relevant for emerging high-resolution crystallographic methods (<1 Å), where conventional density-fitting techniques often fall short in capturing deformation effects[51].

Finally, the ability to quickly compute electronic structures for systems comprising millions of atoms facilitates the generation of large-scale quantum-accurate datasets, an asset for machine-learning applications[52]. By moving beyond small-model systems thanks to the increased computational

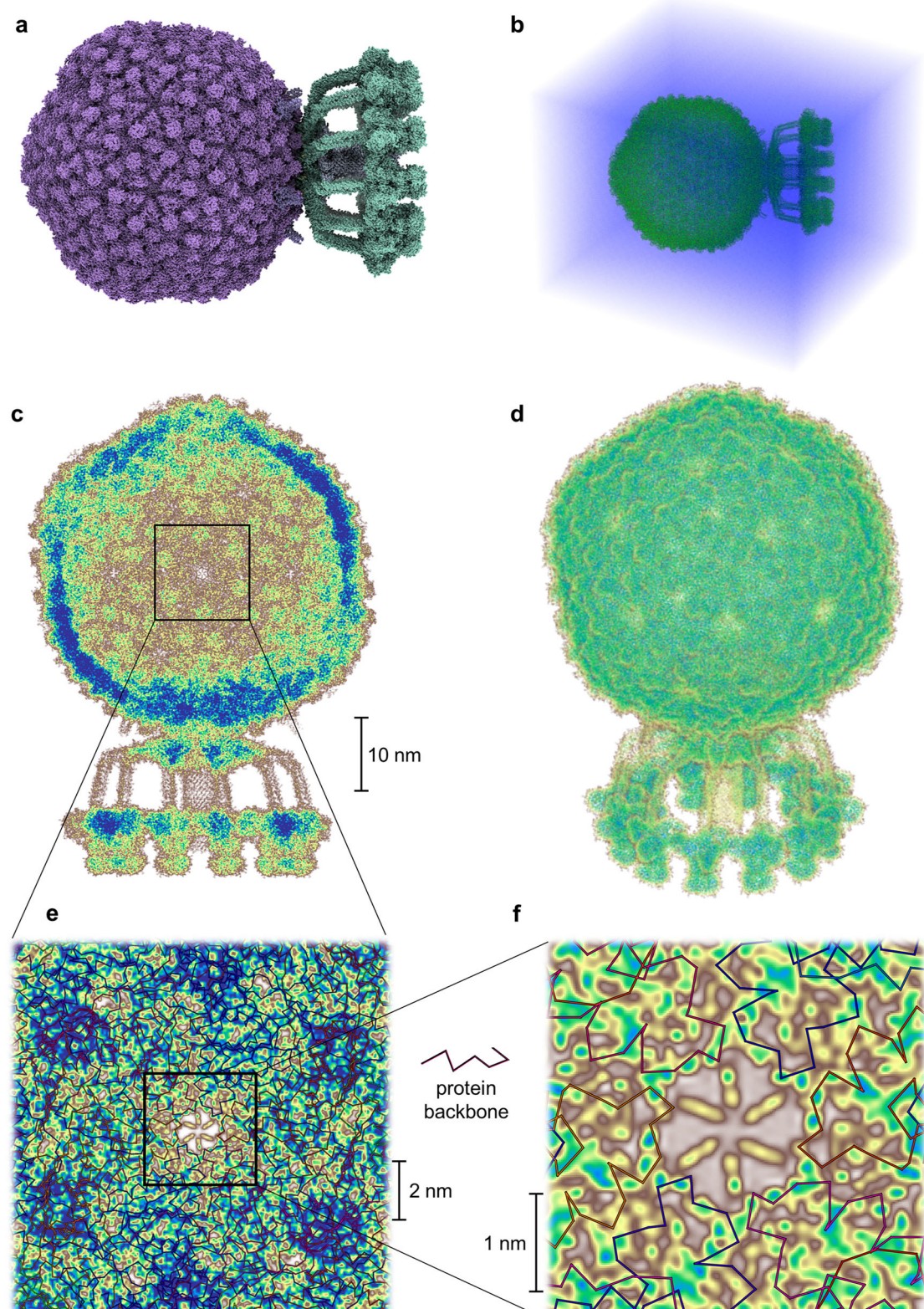

**Fig. 2 | Visualisation of the modified all-electron Hartree–Fock—exemplified on the bacteriophage P68[1] in a solution of water. a, b** Atomic-scale resolved structure of the bacteriophage itself and in solution. **c, d** Calculated electronic density as a projection along the x-axis and as a three-dimensional picture. **e, f** Close-up of the projected electronic density of (**c**) after successive enlargements. Brown corresponds to low and blue to high electronic density values. For a better visualisation the electronic density of the solvent is not displayed and only valence basis function contributions are shown. Visualisation of the protein structure in (**a**) was done with ChimeraX[34].

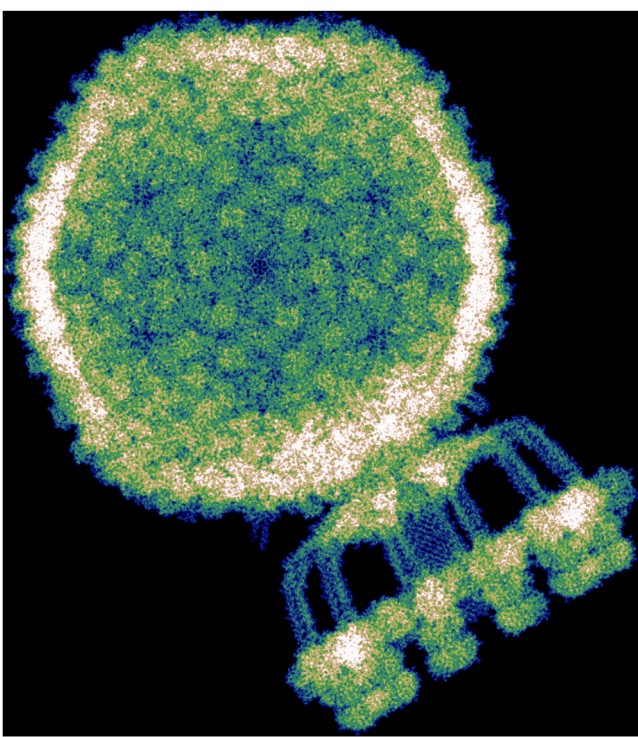

**Fig. 3 | Computed electronic density of the bacteriophage P68.** Detailed visualisation of the electronic density of the bacteriophage P68[1], calculated using the presented Hartree-Fock method. Colours range from black/blue (low density) to white (high density). For improved clarity, the electronic density of the solvent is omitted, and only contributions from valence basis functions are shown.

speed, such datasets may help reduce biases in training and enable broader generalisation in ML-based property predictions.

## Conclusion

Overall, the combination of Hartree–Fock theory with a divide-and-conquer strategy and additional algorithms opens a new regime for large-scale quantum simulations with drastically reduced runtimes and computational resources. In addition to the applications explored here, future directions include fast first-principles molecular dynamics over longer time spans in the range of a few thousand atoms without QM/MM partitioning, calculations under external electric or magnetic fields and description of excited-state or bond-breaking processes. These developments open possibilities in quantum biology, such as investigating the microscopic mechanisms of photosynthesis, understanding the effects of magnetic fields on protein and DNA dynamics, modelling radiation-induced damage at the atomic level, or simulating light-induced conformational changes in chromophoric proteins. Finally, the integration of quantum-accurate datasets with machine-learning techniques may further accelerate the discovery and design of biomolecular functions across biology, medicine and materials science.

## Methods

### Modifications to the Hartree–Fock algorithm

Unlike density functional theory (DFT), Hartree–Fock is a wavefunction-based method and does not directly operate on the electronic density. Each electronic interaction integral contains products of four wave functions. As the system grows, the amount of involved wave functions increases, and the number of electronic repulsion integrals (ERIs) scales with the fourth power of the system size, making their evaluation one of the main computational bottlenecks of Hartree–Fock calculations. To address this, we use systematic approximations that significantly reduce the computational effort. Both wavefunction-wavefunction[35] and density-density interactions are

truncated using physically motivated thresholds. Since wave functions decay exponentially with distance, only nearby orbitals contribute significantly to overlaps (see subsection 'Density cut-offs'). We assess the relevance of a density by calculating the overlap of its two constituent wave functions. For orbital types with negative signs, we use the absolute of the product to capture significant interactions that may otherwise cancel numerically but still produce meaningful contributions, especially to energy gradients, which are important for force calculations. Once densities below the threshold are excluded, the scaling of ERI evaluations drops from quartic to quadratic. To further reduce complexity to linear scaling (order N), we introduce a cut-off for the Coulomb interaction based on inter-density distance (see subsection 'Coulomb cut-offs'). We use a short Coulomb cut-off of 10 Å—a more detailed discussion of this parametrization is found in a separate section. The distance of two densities is defined as the distance between the centres of the two wavefunction pairs that generate them. Only ERIs with Coulomb distances below the Coulomb cut-off are calculated. For ERIs with distances near the cut-off, we apply a smoothing function to ensure continuity.

An ERI is evaluated if the overlap threshold of both densities is exceeded and their centres lie within the Coulomb cut-off. Additionally, we compute the product of the two density relevance values (see the next subsections), divide it by the distance between their centres and compare it to a final relevance threshold. This avoids evaluating interactions between weakly overlapping densities and/or those separated by large distances. To further accelerate the process, we adopt an efficient and parallelizable ERI algorithm[53–55] based on the so-called Gaussian lobe functions[37,56] (see subsection 'Basis function fitting'). Unlike conventional approaches[57–60], which use angular prefactors (x, y, z) to construct directional orbitals, our method builds positive and negative lobes from distinct Gaussian components. This removes angular dependencies and simplifies the integral expressions, making them uniform across orbital types. As a result, all ERI calculations can be performed by using the same expression, which enables a simple parallelisation for both ERIs and their derivatives (necessary to calculate atomic forces). We also precompute components of the ERIs which only depend on pairs of wave functions. This reduces the full calculation of the electronic repulsion between two densities consisting of in total of four Gaussian functions to only 6 additions, 8 multiplications, 10 memory accesses, 1 square root and the evaluation of the error function (see below). For most interactions with large separations, the error function can be approximated as unity, further reducing computational cost.

The following subsections detail the generation of the lobe-based basis functions, the calculation of the density relevance, the calculation of the Coulomb interaction relevance and the mathematical expressions for the modified ERI algorithm.

### Basis function fitting

In electronic structure calculations, wave functions are commonly modelled as Slater-type orbitals (STOs), which for an s-orbital have the form $\varphi_i(\boldsymbol{r}) = A_i \exp\left(-\alpha_i ||\boldsymbol{r} - \boldsymbol{r}_i||\right)$. To enable efficient integration, STOs are typically approximated by using Gaussian functions of the form $g_i(\boldsymbol{r}) = A_i \exp\left(-\alpha_i ||\boldsymbol{r} - \boldsymbol{r}_i||^2\right)$.

Each STO is approximated by a linear combination of $n_G$ Gaussians as

$$\varphi_i(\boldsymbol{r}) = \sum_{a=1}^{n_{\mathrm{G}i}} A_a \exp\left(-\alpha_a ||\boldsymbol{r} - \boldsymbol{r}_a||^2\right), \quad (1)$$

This approximation enables an analytic evaluation of key integrals[57–60], relevant for the overlap matrix, the kinetic and nuclei energy as well as the Coulomb and exchange interaction of the electrons in the system. Orbitals with higher angular momenta are typically expressed via Gaussians containing cartesian prefactors $(x, y, z, x^2, xy, xz, \ldots)$. Although these integrals can also be evaluated analytically, their complexity increases due to the recursive nature of the associated expressions[57–60]. Furthermore, different angular momentum combinations require distinct equations, reducing parallelisation efficiency.

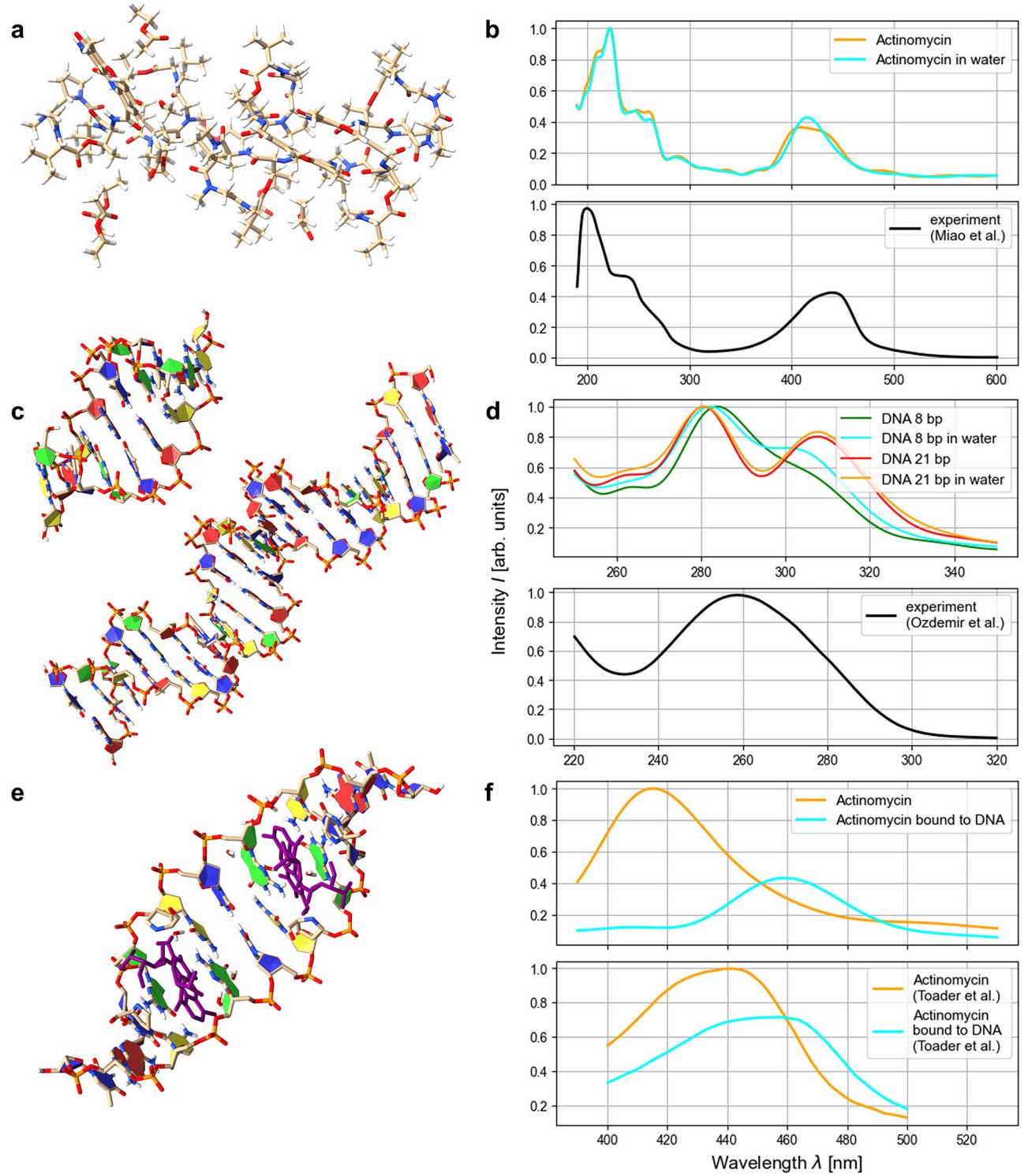

**Fig. 4 | Computed spectra of biomolecules in comparison with experiments.** Absorption spectra were calculated for Actinomycin[5] (**a**), DNA with 8 and 21 base pairs[3,4] (**c**) and Actinomycin bound to DNA[6] (**e**) and compared to experimental data from Miao et al.[42] (**b**), Ozdemir et al.[43] (**d**) and Toader et al.[44] (**f**). Visualisation of the biomolecular structures in (**a**, **c**, **e**) was done with ChimeraX[34].

To avoid this bottleneck, we here adopt Gaussian lobe function expansions[37,56], which allow the construction of basis sets without cartesian prefactors. We show in this work that this approach, though currently not widely used, is particularly well-suited to modern computing architectures optimised for high parallelisation. It also enables the evaluation of ERIs using a unified matrix formalism, which is potentially compatible with GPU

acceleration[61]. The algorithms for ERI evaluation are described in the section 'Electron repulsion integral algorithm'.

For fitting, we start with a Slater-type orbital of the STO-6G type taken from the Basis Set Exchange[62]. In the case of p-orbitals, we approximate them with 6 Gaussian functions, three placed symmetrically slightly to the left and three to the right of the orbital centre. This corresponds in accuracy

| Structure | Hartree–Fock atomic energies | AlphaFold pLDDT score |
|---|---|---|

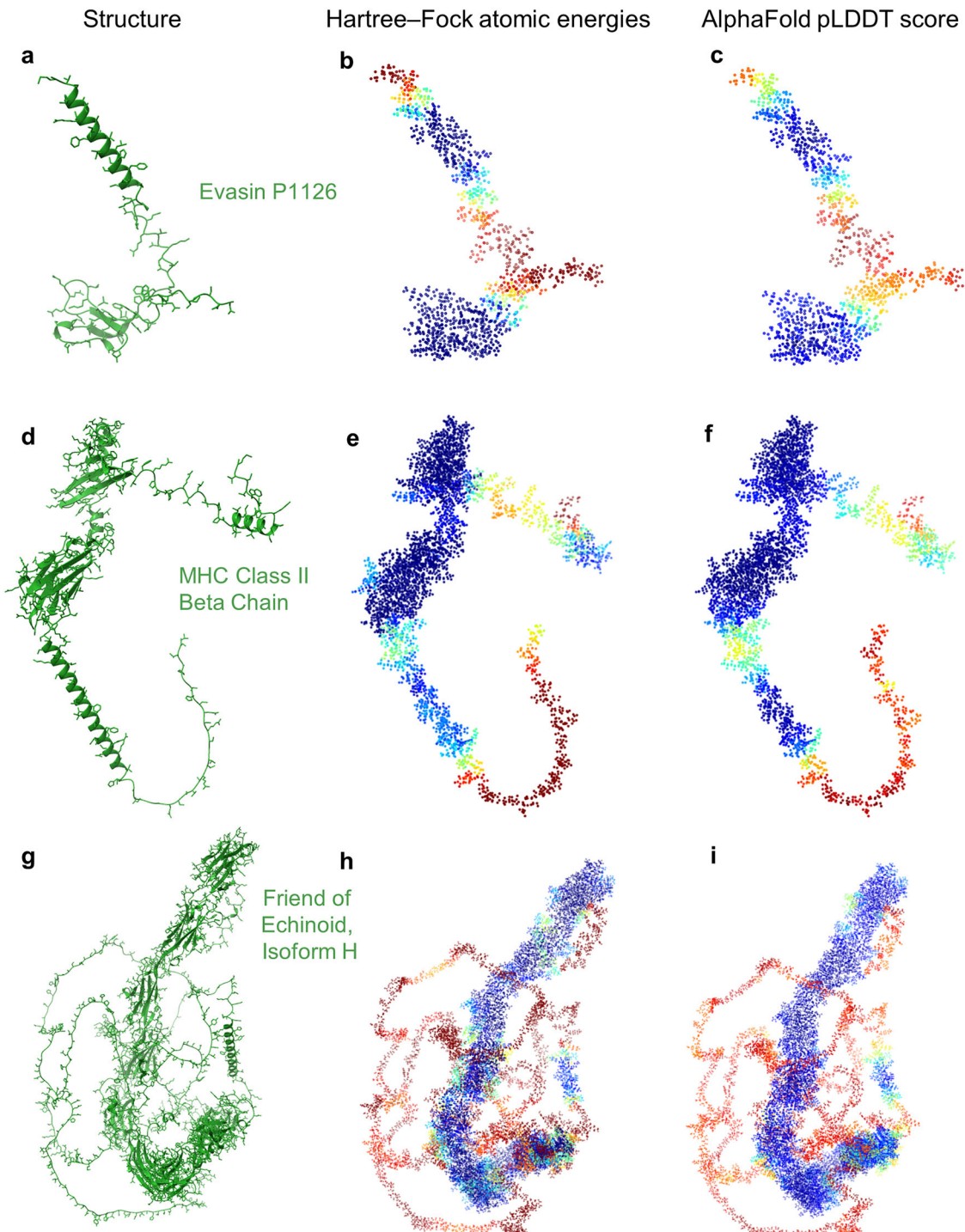

**Fig. 5 | Comparison of quantum-mechanical atomic energies with AlphaFold confidence scores.** Atomic energies from Hartree–Fock calculations are compared with AlphaFold's pLDDT score for three predicted protein structures: Evasin P1126 (PDB AF_AFA0A023FF81F1) (**a**), MHC Class II Beta Chain (AF_A-FA0A023IKK2F1) (**b**) and Friend of Echinoid, Isoform H (AF_AFA0A023GPK8F1) (**c**). For each protein, the left column shows ribbon diagrams and non-hydrogen atoms. The middle column (**b**, **e**, **h**) displays atomic energies computed in a solvated environment (10 Å water shell), smoothed using a Savitzky–Golay filter, scaled by element number and linearly rescaled to match the pLDDT range. The right column (**c**, **f**, **i**) shows the corresponding AlphaFold pLDDT scores mapped onto atoms. Energies of outlier atoms were clipped to match the [0, 100] range of pLDDT values. Protein structures in (**a**, **d**, **g**) were visualised using ChimeraX[34]. Data from rcsb.org[47]. See also Supplementary Fig. 12 for a visualisation using graphs.

to a conventional STO-3G orbital. The fitting is performed on a three-dimensional real space grid. Fit parameters include 3 contraction exponents and 3 coefficients, with the latter multiplied by −1 on one side to model the nodal character of p-orbitals. For the optimisation of the parameters during the fitting process, the trust region algorithm is used. This is done via the function curve_fit from the mathematic Python library scipy[63].

The Gaussian lobe function expansion would also work for orbitals of higher angular momenta. For example, d-orbitals can be split into four lobes

which would require twice as many Gaussians as used for p-orbitals (more for the special case of the $d_{z^2}$ − orbital). The increased computational effort due to the higher number of Gaussian functions should be comparable to the increased complexity of evaluating integrals over Gaussian functions multiplied with cartesian polynomials of degree two in the conventional approaches.

## Density cut-offs

A density $\rho_{ij}(\boldsymbol{r})$ is defined as the product of two wave functions:

$$\rho_{ij}(\boldsymbol{r}) = \varphi_i(\boldsymbol{r})\varphi_j(\boldsymbol{r}) = \sum_{a=1}^{n_{Gi}}\sum_{b=1}^{n_{Gj}} A_a A_b \exp\left(-\alpha_a||\boldsymbol{r} - \boldsymbol{r}_a||^2 - \alpha_b||\boldsymbol{r} - \boldsymbol{r}_b||^2\right).$$

(2)

To determine whether a given density is relevant for further calculations, we compute a scalar value $r\left(\rho_{ij}\right)$, given by the integral over the absolute value of all pairwise products of Gaussian functions:

$$r\left(\rho_{ij}\right) = \int d\boldsymbol{r} \sum_{a=1}^{n_{Gi}}\sum_{b=1}^{n_{Gj}} |A_a||A_b| \exp\left(-\alpha_a||\boldsymbol{r} - \boldsymbol{r}_a||^2 - \alpha_b||\boldsymbol{r} - \boldsymbol{r}_b||^2\right).$$

(3)

This equation is valid for s-type orbitals, where the integrand remains positive. Orbitals with nonzero angular momentum (e.g. p-orbitals) include prefactors $x, y, z$ which would require using the absolute values $|x|, |y|, |z|$ to ensure that the integrand is always positive. However, using these absolute values makes an analytic evaluation of the overlap integrals impossible. To resolve this, we developed an approach to represent each lobe of orbitals with a separate set of Gaussians. The previously described lobe-based basis functions (which achieve a nodal plane by summing Gaussians of different signs) are not suitable here, as their absolute-value integrals do not correspond to the absolute value of the full orbital. Instead, we independently fit each lobe of a p-type orbital using Gaussian functions, and then reconstruct the full orbital by combining lobes with different signs and different centres. The representation of a p-type orbital used for the density relevance calculations is given by:

$$\varphi_i(\boldsymbol{r}) = \sum_{l=1}^{n_{Li}}\sum_{a=1}^{n_{Gi}} s_l A_a$$

$$\cdot \exp\left(-\alpha_{xi,a}\left(x - x_{0i,l,a}\right)^2 - \alpha_{yi,a}\left(y - y_{0i,l,a}\right)^2 - \alpha_{zi,a}\left(z - z_{0i,l,a}\right)^2\right)$$

(4)

where $s_l$ is the sign of lobe $l$ and each cartesian direction has an individual contraction factor (e.g. $\alpha_{xi,a}$) and offset (e.g. $x_{0i,l,a}$) for higher flexibility upon fitting. The fitting process is identical to that described in the section 'Basis function fitting'.

Finally, to determine if a density $\rho_{ij}$ is relevant for the calculation, we compute $r(\rho_{ij})$ and compare it to a threshold value.

## Coulomb cut-offs

In the Hartree–Fock formalism, the Coulomb interaction is modelled as the repulsion between two electronic densities. Following the method described above, Coulomb interactions are neglected if the separation between densities exceeds a defined cut-off. However, since densities are not point-like, a suitable distance metric must be established to estimate their effective separation. For this purpose, each density is assigned an origin, calculated as the arithmetic mean of the atomic positions associated with the two basis functions forming that density. The interaction distance between two densities $\rho_{ab}$ and $\rho_{cd}$ is defined as:

$$d\left(\rho_{ab}, \rho_{cd}\right) = \left|\left|\frac{\boldsymbol{r}_a + \boldsymbol{r}_b}{2} - \frac{\boldsymbol{r}_c + \boldsymbol{r}_d}{2}\right|\right|,$$

(5)

where $\boldsymbol{r}_a, \ldots, \boldsymbol{r}_d$ are the positions of the atoms involved in the four basis functions $a, b, c, d$.

A lower and upper cut-off threshold $r_{c_l}$ and $r_{c_u}$ (typically 8 and 10 Å), are introduced to control the inclusion of interactions. For density pairs with distances between these two thresholds, a smooth transition is applied using a cut-off function to avoid discontinuities. The scaling function (cut-off function) used is

$$f(x) = 1 + 2x^3 - 3x^2,$$

(6)

which smoothly interpolates between 1 and 0 as $x$ increases in the interval $[0, 1]$. The distance interval $[r_{c_l}, r_{c_u}]$ is linearly mapped to the domain $[0, 1]$ of $f(x)$.

## Electron repulsion integral algorithm

An electron repulsion integral (ERI) describes the interaction between two electronic densities:

$$e_{ijkl} \equiv \left(ij, |, kl\right) = \int\int d\boldsymbol{r} d\boldsymbol{r}' \frac{\varphi_i(\boldsymbol{r})\varphi_j(\boldsymbol{r}) \cdot \varphi_k(\boldsymbol{r}')\varphi_l(\boldsymbol{r}')}{||\boldsymbol{r} - \boldsymbol{r}'||}$$

$$= \sum_{a=1}^{n_{Gi}}\sum_{b=1}^{n_{Gj}}\sum_{c=1}^{n_{Gk}}\sum_{d=1}^{n_{Gl}} \int\int d\boldsymbol{r} d\boldsymbol{r}' \frac{A_a A_b A_c A_d}{||\boldsymbol{r} - \boldsymbol{r}'||}$$

$$\cdot \exp\left(-\alpha_a||\boldsymbol{r} - \boldsymbol{r}_a||^2 - \alpha_b||\boldsymbol{r} - \boldsymbol{r}_b||^2 - \alpha_c||\boldsymbol{r}' - \boldsymbol{r}_c||^2 - \alpha_d||\boldsymbol{r}' - \boldsymbol{r}_d||^2\right).$$

$$\equiv \sum_{a=1}^{n_{Gi}}\sum_{b=1}^{n_{Gj}}\sum_{c=1}^{n_{Gk}}\sum_{d=1}^{n_{Gl}} \left(g_a g_b | g_c g_d\right),$$

(7)

where each orbital is expanded in terms of Gaussian functions. Each ERI of the individual Gaussian functions is given by:

$$\left(g_a g_b | g_c g_d\right) = O_{g_a g_b} O_{g_c g_d} P_{g_a g_b g_c g_d}\, {}_1F_1\left(\frac{1}{2}, \frac{3}{2}, -P_{g_a g_b g_c g_d} d_{g_a g_b g_c g_d}^2\right)$$

(8)

with the confluent hypergeometric function $_1F_1$, two overlaps $O_{g_a g_b}$ and $O_{g_c g_d}$, a factor $P_{g_a g_b g_c g_d}$ and a distance $d_{g_a g_b g_c g_d}$ which are defined as:

$$O_{g_a g_b} := \sqrt{2}\pi^{\frac{5}{4}} A_a A_b \frac{1}{\left(\alpha_a + \alpha_b\right)^{\frac{3}{2}}} \cdot \exp\left(-\frac{\alpha_a \alpha_b}{\alpha_a + \alpha_b}||\boldsymbol{r}_a - \boldsymbol{r}_b||^2\right),$$

(9)

$$P_{g_a g_b g_c g_d} := \frac{1}{\frac{1}{\alpha_a + \alpha_b} + \frac{1}{\alpha_c + \alpha_d}},$$

(10)

$$d_{g_a g_b g_c g_d} := ||\boldsymbol{r}_{ab} - \boldsymbol{r}_{cd}||, \boldsymbol{r}_{ab} = \frac{\alpha_a \boldsymbol{r}_a + \alpha_b \boldsymbol{r}_b}{\alpha_a + \alpha_b}.$$

(11)

We define $x := P_{g_a g_b g_c g_d} d_{g_a g_b g_c g_d}^2$ and evaluate the confluent hypergeometric function using the following equation:

$$_1F_1\left(\tfrac{1}{2}, \tfrac{3}{2}, -x\right) = \sqrt{\pi}\frac{\mathrm{erf}(\sqrt{x})}{2\sqrt{x}},$$

(12)

where the error function can be efficiently approximated using:

$$\mathrm{erf}(x) \cong 1 - \left(c_1 t + c_2 t^2 + c_3 t^3 + c_4 t^4 + c_5 t^5\right) \cdot \exp\left(-x^2\right), t = \frac{1}{1 + c_0 x}$$

(13)

(with coefficients $c_0 = 0.3275911$, $c_1 = 0.254829592$, $c_2 = 0.284496736$, $c_3 = 1.421413741$, $c_4 = -1.453152027$ and $c_5 = 1.061405429$)[64]. For large $x$ the error function $\mathrm{erf}(x)$ rapidly approaches 1, since $1 - \mathrm{erf}(4) \approx$

$1.5 \cdot 10^{-8}$ and $1 - \mathrm{erf}(5) \approx 1.5 \cdot 10^{-12}$, and can be approximated accordingly.

We recall that we evaluate the ERIs of the form

$$(ij|kl) = \sum_{a=1}^{n_{Gi}} \sum_{b=1}^{n_{Gj}} \sum_{c=1}^{n_{Gk}} \sum_{d=1}^{n_{Gl}} (g_a g_b | g_c g_d). \tag{14}$$

It will only be necessary to compute a small part of all possible combinations $(ij, |, kl)$, since we use both a density cut-off and a Coulomb cut-off. Therefore, and to reduce redundant computation, the quadruple summation over all Gaussians is reformulated as a double summation over precomputed Gaussian pairs $g_a g_b$ and $g_c g_d$, for which the quantities $O_{g_a g_b}$ and $\boldsymbol{r}_{ab}$ as well as $O_{g_c g_d}$ and $\boldsymbol{r}_{cd}$, respectively, are stored in lookup tables. This is practical because number of density pairs is considerably smaller than the total number of ERIs. The complete ERI evaluation algorithm can be visualised in terms of a three-dimensional tensor: the first and second axis are the sums described above and the third axis is the list of all relevant combinations $(ij, |, kl)$. Summing up this tensor along the first two axes yields the full list of ERIs. It must be noted that each integral $(g_a g_b | g_c g_d)$ in this tensor is computed by using the same equation. However, the lengths of the first two axes $n_{Gi} n_{Gj}$ and $n_{Gk} n_{Gl}$ are not always the same as the number of used Gaussian functions is not the same for all wave functions.

To account for this, we modify the equation for $(ij|kl)$ by setting the limits of the sums to max $n_{Gi} n_{Gj}$ and max $n_{Gk} n_{Gl}$ which leads to:

$$(ij|kl) = \sum_{a,b=1}^{\max n_{Gi} n_{Gj}} \sum_{c,d=1}^{\max n_{Gk} n_{Gl}} \delta^{ijkl}_{g_a g_b g_c g_d} (g_a g_b | g_c g_d). \tag{15}$$

where $\delta^{ijkl}_{g_a g_b g_c g_d}$ is 1 if $ab \leq n_{Gi} n_{Gj}$ and $cd \leq n_{Gk} n_{Gl}$—otherwise it is 0. Therefore, $\delta^{ijkl}_{g_a g_b g_c g_d}$ describes if the index combination $abcd$ is defined for the set of Gaussian functions $g_a g_b g_c g_d$ within $(ij, |, kl)$.

This method allows the evaluation of all $(ij|kl)$ with the same equation and is therefore highly parallelizable. Additionally, it saves memory, since no intermediate values are stored (all Gaussian ERIs $(g_a g_b | g_c g_d)$ become contracted immediately in each step of the loop over $(ij|kl)$). Precomputing and storing intermediate values dependent on only one density ($O_{g_a g_b}$, $O_{g_c g_d}$, $\boldsymbol{r}_{ab}$ and $\boldsymbol{r}_{cd}$) reduce the computational effort as well.

## Implementation check
To ensure that the Hartree-Fock implementation runs correctly, especially under the implemented cut-off methods and basis function manipulations, results were benchmarked against the quantum-chemistry package ORCA[13,65–69]. Specifically, we assessed the agreement between our results and a standard calculation using the STO-3G basis set, focusing on both total and orbital energies. This allowed us to ensure that the programme runs correctly and that lobe function expansion and interaction truncations do not lead to strongly deviating results. As a test system, we selected the molecule beta-carotene ($C_{40}H_{56}$), which involves 541,089,856 unique ERIs, of which 30,572,290 were considered to have relevant density contributions. Among these, 3,946,243 integrals met both density and Coulomb interaction relevance criteria.

The total electronic energy calculated using our implementation was −1528.547 hartree (Ha), compared with −1528.663 Ha obtained via ORCA. This corresponds to an absolute deviation of 0.115 Ha, or 0.0075%. For the orbital energies, we compared all valence orbitals and the ten lowest virtual orbitals, yielding a root mean square deviation of 7.98 mHa. Using only the Coulomb cut-off leads to a value of 7.57 mHa, whereas only density cut-offs result in an error of 6.07 mHa. No cut-offs at all result in a deviation of 3.23 mHa which is attributed to the modified basis functions in the lobe function expansion which do not match the STO-3G basis exactly and therefore lead to a small error.

In terms of computational performance, ORCA required 192 s for a beta-carotene calculation on an Intel i5 processor. In comparison, our implementation with all cut-offs was completed in 21 s. When disabling the Coulomb cut-off and using a minimal density threshold of $10^{-10}$, typical of many Hartree–Fock codes, the runtime increased to 131 s.

## Parallel Hartree–Fock on large systems
The density cut-off was set to $10^{-4}$ and the Coulomb cut-off to 10 Å with a smooth transition to zero starting at 8 Å. The self-consistent cycle of the Hartree–Fock calculation used a convergence threshold of $10^{-6}$ for the RMSD of the density matrices. The STO-3G basis set was split into individual Gaussian functions to allow an ERI evaluation according to the above-mentioned Gaussian lobe function expansion algorithm.

It has to be kept in mind that the short Coulomb cut-off neglects long-range interactions since no PME algorithm or similar methods are used. In the divide-and-conquer approach this is forced to guarantee a subsequent evaluation of the clusters. For the applications of UV/Vis spectra and atomic energies, this truncation only has a small to negligible influence on the results. Other properties are also inaccessible if the divide-and-conquer approach is used: these include the system-wide molecular orbitals such as for example HOMO and LUMO and the band gap. Orthogonality between the divide-and-conquer regions is also not maintained, the approach is no longer variational and there would be no access to MD simulations. Note that for applications up to a few thousand atoms, the formalism runs well without the divide-and-conquer approach which makes the properties mentioned above accessible in this region.

System preparation for the large-scale calculations involved solvation in water using the solvation tool of the programme Visual Molecular Dynamics (VMD)[70]. The solvated system was then partitioned into subsystems on a 3D mesh. Hartree-Fock calculations were carried out independently for each subsystem. To reconstruct the total electronic density, the density matrix elements were chosen from the subsystem containing both atoms corresponding to the two basis functions. If the basis functions were associated with different subsystems, the mean of the relevant density matrix elements was used.

The total electronic density was then calculated as

$$\rho_{\text{total}}(\boldsymbol{r}) = \sum_{a=1}^{n_{\text{bf}}} \sum_{b=1}^{n_{\text{bf}}} r(a,b) P_{ab} \rho_{ab}(\boldsymbol{r}), \tag{16}$$

where $n_{\text{bf}}$ is the total number of basis functions, $r(a,b)$ is a relevance determining whether the pair basis functions $a$ and $b$ contributes significantly, and $P_{ab}$ is the corresponding element of the density matrix.

Two methods were used for the interpolation of density matrix elements. One method was to compute the new density element as the mean of the density elements of the clusters that get merged. In the current and more accurate method, which is the one found in the source code, the cross-section between the line connecting the two atoms involved in one density and the border between the two involved clusters is computed. This yields two new lengths—between the cross-section point and the two atoms—which are normalised to a sum of 1 and serve as the prefactors for the density contributions from the two subsystems. The length in one subsystem serves as the prefactor for the density of the other subsystem, for example if one atom is close to the border, then the contribution of the other subsystem, where the atom is further away from the border, will be small. The described atoms are the atoms associated with the two basis functions that each density is composed of.

## Spectral calculation parameters
For the quantum spectral calculations, we employed the RT-TDHF formalism, as it scales more favourably for larger systems than the linear-response time-dependent Hartree–Fock (LR-TDHF) approach, which theoretically scales with the sixth power of the system size.

In RT-TDHF the system is excited by an electric field pulse and propagated in time using quantum-mechanical time evolution. During the

propagation, the dipole moments are recorded at each time step and the frequency-dependent polarizability and absorption spectra are obtained via discrete Fourier transform of the dipole time series. The systems were propagated over 2000 steps with a time step of 0.25 atomic units (a.u.), corresponding to a total propagation time of 500 a.u. (12.1 fs). The applied electric field pulse had a strength of $10^{-5}$ a.u., a standard deviation of 0.2 a.u. and a starting time of 2 a.u. During the Fourier transform, an attenuation factor of 0.01 a.u. was used to supress numerical noise.

Following established practice, the resulting spectra were rescaled[71,72] to account for the systematic shifts in excitation energies inherent to time-dependent Hartree–Fock and density functional theory calculations. A rescaling factor of 1.335 was applied, consistent with the benchmark study by Jacquemin et al., who computed TDHF spectra for a wide range of organic dyes[72].

For the spectral calculations, the density cut-off was chosen as $10^{-6}$ and the Coulomb cut-off as 10 Å. This parametrization lead to results in good agreement with experimental data and in a test showed only small deviations from a calculation using no cut-offs. This test and calculations with further reduction of the density or Coulomb cut-off are shown in Supplementary Fig. 11. A density cut-off of $10^{-4}$ showed similar results to a value of $10^{-6}$ in this test but lead to some absorption peaks vanishing or decreasing in intensity in other tests with benzene and DNA, therefore we chose a cut-off of $10^{-6}$ as the standard parametrization. For the Coulomb cut-off 8 Å also lead to similar results to 10 Å—we nonetheless use 10 Å as the default Coulomb cut-off to have some buffer as the stability might differ from system and system and to not discard too many mid-range interactions.

## Atomic energy processing

The atomic energies used for the comparison with AlphaFold's pLDDT scores were smoothed using a Savitzky-Golay filter[73] with a window length of 150. To match the scale of pLDDT values, the computed energies were multiplied by the factor of 15 and shifted by a structure-specific offset (–430, –440 and –460 for PDB entries AF_AFA0A023FF81F1[47], PDB AF_AFA0A023IKK2F1[47] and AF_AFA0A023GPK8F1[47], respectively). The rescaled energies were truncated at the corresponding structure's maximum and minimum pLDDT values to eliminate extreme atomic energy outliers.

A modified expression for atomic energies was employed, defined as

$$E_a = \frac{1}{Z_a} \sum_{i=1}^{n_{\mathrm{bfv}a}} \sum_{j=1}^{n_{\mathrm{bf}}} P_{ij}\left(H_{ij} + \tfrac{1}{2}G_{ij}\right), \tag{17}$$

where $E_a$ is the atomic energy of atom $a$, $Z_a$ is the atomic charge (used for normalisation), $n_{\mathrm{bfv}a}$ is the number valence basis functions for atom $a$ (excluding core contributions), and $n_{\mathrm{bf}}$ is the total number of basis functions. $P_{ij}$, $H_{ij}$ and $G_{ij}$ denote the density matrix, core Hamiltonian and two-electron interaction matrix elements, respectively.

System preparation was based on the aforementioned PDB entries[47], with solvation performed using a 10 Å water shell generated via the solvation tool of Chimera[74].

The actual and predicted pLDDT scores for the three investigated structures are shown as graphs in Supplementary Fig. 12.

The accuracy of subsystem splitting was tested using the protein Evasin P1126 (PDB ID: AF_AFA0A023FF81F1)[47]. For atomic energy calculations, only valence electrons were considered. Subsystems were constructed by segmenting the amino acid chain of the whole protein into smaller peptide fragments. Comparing a full Hartree-Fock calculation with the subsystem-based approach, we obtained a root mean square deviation of 0.385 Ha for the atomic energies, corresponding to a relative error of 0.444% and a Pearson correlation coefficient of 0.9986. This is shown in Supplementary Fig. 13.

## Programme implementation

The described programme is written in the Python programming language[75]. To achieve high computational performance, all computational demanding functions are written either in matrix form or compiled to machine code with just-in-time (JIT) compilation. For JIT compilation, the library Numba[76] was used, and matrix operations were performed with Numpy[77] or PyTorch[78]. Visualisation of results and plotting were performed using the library Matplotlib[79].

## Biomolecular structures and visualisation

All calculations in this article were based on biomolecular structures obtained from the RCSB Protein Data Bank (PDB)[47]. The structures and their full references (including original research articles and PDB entries) are as follows: PDB 2M3X[33] (12-bladed propeller protein), PDB 1H68[36,80] (rhodopsin), PDB 6Q3G[1,81] (bacteriophage P68), PDB 1A7Y[5,82] (Actinomycin D), PDB 1D79[3,83] (DNA with 8 base pairs), PDB 2JYK[4,84] (DNA with 21 base pairs), PDB 1MNV[6,85] (Actinomycin D bound to DNA), PDB 8WL2[38,86] (flagellar motor), PDB 7PKR[39,87] (vault protein complex). Structures used for comparing AlphaFold's pLDDT scores to atomic energies include: PDB AF_AFA0A023FF81F1 (Evasin P1126), PDB AF_AFA0A023IKK2F1 (MHC Class II Beta Chain) and PDB AF_AFA0A023GPK8F1 (Friend of Echinoid, Isoform H). These were obtained from the Computed Structure Models (CSM) section of the RCSB PDB[47].

Visualisations of biomolecular structures were created using ChimeraX[34,74,88,89] and the PDB Mol* Viewer[90] for Supplementary Figure 10. All other figure elements (including biomolecular structures at atomic resolution, electronic densities, absorption spectra and atomic energies) were generated with Matplotlib[79] for python.

## Further testing with molecular dynamics

As a test system to study structural optimisation and molecular dynamics, we used a DNA tetramer (sequence ACGT). We calculated the forces for this structure with and without using the Coulomb cut-off. The parametrization of the Coulomb cut-off is the same for the electron repulsion integrals (ERIs) and for the ion-ion interactions. The result is shown in Supplementary Fig. 14.

In addition to this, a structural optimisation was performed for the molecule. For this the Broyden–Fletcher–Goldfarb–Shanno (BFGS) algorithm was used—a quasi-Newton method which uses an approximation of the Hessian and which is commonly used for structural optimisation or for nonlinear optimisation problems in general. Compared to gradient descent methods which do not precondition the gradient vector, the BFGS algorithm usually shows much faster convergence. However, a convergence to zero or values very close to zero is usually not achieved for larger molecules[16] since this would also require to perform a structural optimisation at a larger scale. Therefore, the root-mean-square (RMS) of the forces is often improved by one or two orders of magnitude until the algorithm plateaus.

The development of the RMS of the forces during a BFGS optimisation of the ACGT molecule is shown in Supplementary Fig. 15 for two calculations without and with Coulomb cut-off (cut-off values at 3.5 and 10 Angstrom), respectively. The calculation was run over 30 optimisation steps. In each iteration, the optimal descent prefactor was determined using a line search utilising the golden section search algorithm which avoids re-evaluations as much as possible. During the line search, only the energy is needed and not the forces and therefore the programme presented in this work was run in energy-only mode to save computation time. In Supplementary Fig. 15 we observe convergence for the calculations with and without cut-off. Both converge in a very similar way and show oscillatory convergence at the end. The minimum RMS is achieved at step 28 with a value of around 0.0009 for both calculations.

In Supplementary Fig. 16 the atomic coordinates from before and after the structure optimisation are shown as an overlay. Additionally, at the bottom, the final coordinates from the structural optimisation with Coulomb cut-off are overlayed with the final coordinates from the optimisation without Coulomb cut-off. Here, we observe that the structural optimisation does have an influence on the geometry of the system (see top panel of Supplementary Fig. 16), as it is expected. Furthermore, the final geometry is not significantly influenced by the Coulomb cut-off as both calculations lead

to a very similar result. The root-mean-square displacement between the two calculations is computed as 0.047 Angstrom.

It should also be mentioned that forces that deviate from the forces without cut-offs (both density and Coulomb cut-off) still are the derivative of a (now slightly changed) energy landscape. Nonetheless, the sum of the forces still remains zero as they are the exact derivative of the energy. This has been tested by ensuring that the forces obtained from the programme can be reproduced numerically as the limit case of the finite-differences method. Testing has been done for the forces as a sum of derivatives as well as for the derivatives of the integrals itself (overlap, kinetic, nuclei and electron repulsion integrals).

We continue the testing by investigating the stability of molecular dynamics simulations with and without the Coulomb cut-off. To do so, we performed an MD simulation of the same system that was used above for the investigation of the structural optimisation process. The simulation runs over 200 time steps with a step interval of 0.2 fs.

It has to be mentioned that a slight drift in the total energy can be seen with a value of 6.0e-7 Ha per atom and per femtosecond, this is caused by some of the starting forces being very large and is not an error which occurs due to cut-offs as it happens in the calculation without cut-offs as well. The MD simulation was done with large forces to test the performance of the cut-offs for more extreme situations beyond the equilibrium—for this reason, we also chose to work with the rather drastic cut-off scheme 3.5/10. However, to ensure that drift of the total energy is indeed caused only by the integrator, a simulation was done with a smaller time step for a smaller system (DNA with just one base pair, cytosine) with the Coulomb cut-off used. In this case it could be observed that the changes of the total energy can be reduced by decreasing the time step of the molecular dynamics simulation. To be precise, reducing the time step ten-fold leads to an energy conservation error of 8.0e-8 Ha per atom and per femtosecond. For an MD simulation with 10 steps of geometry optimisation prior to its start, a time step of 0.5 fs can be used with an energy conservation error of 2.9e-9 Ha/atom/fs.

In total, we observe very little difference between the kinetics in the two MD simulations with and without cut-off used. Of course, this is not a full test, therefore we also investigate the actual trajectories. However, from Supplementary Fig. 17 it can be seen that the low Coulomb cut-off does not lead to numerical instabilities or to larger forces than in the simulation with cut-off.

Next, we consider the trajectories obtained in the two simulations. We computed the RMSD between the coordinates of the two simulations at each time step. This is shown in Supplementary Fig. 18 below as the green line. Additionally, the mean travelled distance of the atoms in the simulation was computed for both simulations is also displayed in Supplementary Fig. 18.

Here, we see a difference in the trajectories of the two MD simulations but it is small compared to the total movement of the atoms. The maximum RMSD is 0.091 Angstrom. In Supplementary Fig. 19 the atomic coordinates from the start and the end of the MD simulation with no cut-off are over-layed and a second overlay is shown where the final MD coordinates of the simulation with no cut-off are compared with the final coordinates of the MD simulation with the Coulomb cut-off.

We observe again that the changes due to the cut-off are small compared to the total movement in the MD simulation. Note that here a quite aggressive cut-off configuration (3.5 and 10 Angstrom for the lower and upper cut-off) is used which is not recommended due the very small value of 3.5 Angstrom for the lower cut-off. In the following paragraph we discuss the reason for the large deviation between the lower and upper cut-off and why a higher cut-off value is recommended for force calculations.

The choice of the lower and upper cut-off plays an important role, specifically for force calculations. For the calculations that only compute the energy of the system, the difference between the two cut-offs only has the purpose to smoothen the transition to zero. However, if the energy gradient is computed, a fast transition from the actual value of the Coulomb interaction to zero can lead to very large derivatives. This can lead to the derivative between the two cut-off values being larger that the derivative for

distances below the lower cut-off value. Such a behaviour has to be avoided to maintain a smooth transition to zero for the forces as well. This is achieved by increasing the distance between the values of the two Coulomb cut-offs, ideally the upper Coulomb cut-off should be at least 2.5 times larger than the lower cut-off to ensure a smooth transition to zero for the forces. For this reason, a cut-off configuration of 8/20 is more recommended which was tested at the beginning of this section for the force comparisons.

To investigate the effect of the divide-and-conquer method on atomic forces, we performed the same test as for the Coulomb cut-off where the forces are computed with and without the approximation used. Here, we study a larger system in the form of the DNA 12-mer ACGTACGTACGT which will be split into three separate sections during the divide-and-conquer formalism. We compute the forces using the Coulomb cut-off configuration (3.5/10) used for the investigation of the effects of the Coulomb cut-off.

Two different configurations are tested, one with a buffer zone of 8 and one with 10 Angstrom. Neighbouring atoms to the buffer zone and hydrogen atoms for bond termination have been added as described in the respective section in the main text.

From Supplementary Fig. 20 we can see that the errors of the forces are quite low, even for the smaller buffer zone at 8 Angstrom. One explanation for this is that the Coulomb cut-off already truncated a lot of the long-range interaction but nonetheless this test gives the information that no significant additional errors occur on top of those caused by the Coulomb cut-off. Furthermore, we see that the distribution of the electron density in the core region is not significantly influenced by the parts not included in the buffer zone because otherwise stronger deviations in the forces would be expected. Another explanation for the low error is the adding of neighbouring atoms to avoid breaking bonds and the replacement of heavier atoms with hydrogen atoms at the section borders which help to construct physically sensible boundary conditions, potentially reducing errors further.

For investigating interactions between molecules, we consider the binding energies between two molecules and two proteins as two test systems. To be precise, the two systems studied are the vitamin biotin (vitamin H) bound to the protein avidin (PDB 1AVD)[91,92] and the drug FK506 bound to the protein FKBP (PDB 1FKJ).[93,94] Both of these systems have high binding energies of −20.4 and −12.8 kcal/mol, respectively.[95,96] Since the computation of binding energies can be considered an area where high accuracies are necessary, it serves as a test if minimal-basis Hartree-Fock can still provide somewhat reliable results.

For this test we perform a single-point energy calculation using Hartree-Fock and try out different cut-off configurations, with no divide-and-conquer used. Since the whole protein with ligand is too large to study without cut-offs, we focus on the area around the ligand. Therefore, the system used in the calculations consists of the ligand and the environment within 4 Angstrom around the ligand. Additionally, bond cutting is accounted for by adding further atoms and replacing some atoms with hydrogen atoms to terminate bonds. The exact procedure is described in the corresponding section in the main text and the resulting systems as well as the complete protein-ligand complex are shown in Supplementary Fig. 21.

For the two systems discussed above, the binding energy is computed as the energy difference between the energy of the protein-ligand complex and the sum of energies of the protein and the ligand (with 'protein' we here refer to the part of the protein treated in the calculation, as described above). In Supplementary Table 1 the resulting energies are listed for different combinations of density and Coulomb cut-offs.

Here, we see limitations due to the use of cut-offs. Especially for the Coulomb cut-off configurations 3.5/10 and 8/10, the binding energy shows strong deviations from the results with no Coulomb cut-off—in some cases no binding is predicted. A reduced density cut-off of 1.0e-5 also show a deviation for the biotin/avidin complex. A higher Coulomb cut-off configuration (15/20) shows less strong deviation from the binding energy without a Coulomb cut-off. Lowering the density cut-off to values of $10^{-7}$ or $10^{-8}$ has little influence on the computed binding energy. However, for the

calculations with the highest accuracy, the experimental binding energies can be reproduced with errors of 2.3 and 1.2 kcal/mol.

These results show that using low Coulomb cut-offs makes it impossible to use the formalism for applications which need a higher accuracy. In these cases, it is necessary to use no (or a much larger) Coulomb cut-off. The density cut-off at $10^{-6}$ appears to be a reasonable choice which should not compromise the accuracy significantly. Consequently, testing is required if a Coulomb cut-off or a higher density cut-off is used for applications beyond those studied in this work to ensure that the accuracy is sufficient for the investigated problems. If no Coulomb cut-off is used and a density cut-off smaller than $10^{-6}$ is employed, we observe that we can reproduce the experimental binding energies for the two studied systems within a reasonable error.

## Data availability

Data is deposited at Zenodo (https://doi.org/10.5281/zenodo.16889519). https://doi.org/10.5281/zenodo.16889518.

## Code availability

Code is deposited at Zenodo (https://doi.org/10.5281/zenodo.16889519). https://doi.org/10.5281/zenodo.16889518.

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

## Acknowledgements

Both authors acknowledge support from the Deutsche Forschungsgemeinschaft through the RTG 2749/1: "Biological Clocks on Multiple Time Scales". The authors gratefully acknowledge the computing time provided to them on the high-performance computer Lichtenberg II at TU Darmstadt, part of the National High Performance Computing Center for Computational Engineering Science (NHR4CES). Calculations were also performed on the Linux Cluster of the University of Kassel and a cluster of the research group. The authors thank Pedro Ojeda-May for useful discussions. Biomolecular graphics were performed with UCSF ChimeraX, developed by the Resource for Biocomputing, Visualisation and Informatics at the University of California, San Francisco, with support from National Institutes of Health R01-GM129325 and the Office of Cyber Infrastructure and Computational Biology, National Institute of Allergy and Infectious Diseases. L.W. and M.E.G. disclose support for the research of this work from the German Research Association through the Research Training Group "Multiscale Clocks" (RTG 2749/1).

## Author contributions

L.W. initiated the project and developed the numerical method and the computer code. M.E.G. determined the application fields of the method and supervised the project. L.W. performed the calculations. Both authors analysed the results and wrote the final version of the manuscript.

## Funding

## Competing interests

The authors declare no competing interests.
