## [Transparent Peer Review file · Communications Chemistry]

A Quantum-Mechanical Framework for Million-Atom Scale Biological Systems

Corresponding Author: Professor Martin Garcia

This manuscript has been previously reviewed at another journal. This document only contains information relating to versions considered at Communications Chemistry.

Version 0:

Reviewer comments:

Reviewer #1

(Remarks to the Author)

The authors' responses to the referees are helpful and clear, but I am still not sure whether the paper is ready for publication (and some responses have identified new areas of concern). While the scale of calculations is large, it is not clear how reliable they are, nor how widely applicable the approach will be. I give detailed comments below.

1. The authors have performed no structural optimisation or molecular dynamics in the present paper. This is an important issue because I am concerned that their truncated ion-ion interaction will lead to wrong forces, and hence wrong structures or dynamics. Similarly, the lack of dispersion at even an empirical level will have an effect. Ideally this would be tested, but it certainly must be highlighted.
2. The lack of communication between subsystems in the divide and conquer method (and the lack of total electron number conservation that the authors allude to) strikes me as a significant issue, particularly for structures which are charged or polarised, and might well affect optimisation or dynamics. Some form of testing would be helpful.
3. I am concerned that Figure 5, one of their key results, is not a sensitive test of this method (as indicated by Referee 3 in their report). No colour map is given, and it is very hard to get any quantitative measure of deviation by comparing visually like this.
4. Any interactions between molecules (as suggested in the first sentence of the introduction) will be strongly affected by the types of problem I suggest above. Since a major application of this method seems to be this kind of question, some characterisation of the limitations is needed.

Version 1:

Reviewer comments:

Reviewer #1

(Remarks to the Author)

I would commend the authors for their extensive work to check the accuracy of their method. I was a little surprised at how insensitive the simple molecular relaxations and dynamics were to the cutoffs, but this is useful data.

However, the results in Table 1 in the rebuttal letter and the extended data highlight my key concern with the implementation of divide and conquer. I disagree with the authors' interpretation: I do not think that they can claim any kind of convergence here with the Coulomb cutoff - the oscillations in binding energy are still significant. I think that there should be some discussion of this in the text.

Moreover, I think that some of the claims made in the abstract and introduction need to be removed or toned down; in particular in lines 29-31 ("This approach opens new avenues in quantum physics, structural biology, spectroscopy, bioinformatics, medicine, and materials science.") and lines 50-53 ("This advance bridges quantum mechanics and biology

at a very high computational speed, enabling large-scale, first-principles simulations with broad applications in quantum biology, structural biology, medicine, materials science, and many-body physics.") I do not think that the present results justify such wide-ranging claims: in particular it's not clear that molecule-molecule interactions are reliable with the Coulomb cutoff, and there is no evidence presented on the efficacy of the method in materials science or related areas.

Reply letter

“Quantum Mechanics at the Million-Atom Scale: From Viruses to Protein Folding”

Luc Wieners, Martin E. Garcia

Introduction:

We thank the reviewer for their comments. Furthermore, we can of course provide more detailed tests of the presented method and agree that this will help to support the reliability of the framework for the applications in this work.

For point 1 and 2 regarding the truncated ion-ion cut-off, the divide-and-conquer approach and the investigation of structural optimization and molecular dynamics, we can certainly provide test results. In fact, the Coulomb cut-off was tested on its influence on force calculations as forces were implemented in a previous version of the presented code. While they are not available in version 3.10 which is the code attached to this work, forces are supported again in more recent versions. Therefore, we can provide results for structural optimizations and MD simulations for validation purposes.

Point 3 is discussed in a supplementary figure which provides a quantitative representation for the obtained results. For point 4 we present calculations that show that interactions between molecules can be computed reliably with this framework and also discuss limitations of the method.

In addition to this, we here want to point out that the presented framework and its validation is tailored for the applications presented in this work. Therefore, we do not want to make claims regarding the suitability of the method for applications beyond the presented results in this work. This would of course need further validation, and we want to keep the focus of this work on the spectral calculations and the quantum-mechanical computation of confidence scores, both of which are tested against external results. While the scope of the work was probably miscommunicated in the first submission, we hope that this is now expressed clearly. Therefore, we – for example – consider molecular dynamics simulations outside the scope of this work. Nonetheless, we added some simulations, since we agree that they can provide further validation of the method.

Point-by-point response to the referee’s comments:

Reviewer #1 (Remarks to the Author):

“The authors' responses to the referees are helpful and clear, but I am still not sure whether the paper is ready for publication (and some responses have identified new areas of concern). While the scale of calculations is large, it is not clear how reliable they are, nor how widely applicable the approach will be. I give detailed comments below.”

In the following, we provide validations of the method in form of computing forces with and without cut-offs as well as forces with and without the using the divide-and-conquer

approach. Additionally, a structural optimization and a molecular dynamics simulation is performed with and without using the cut-off which show very little deviation and support the validity of the presented method. Computed confidence scores calculated with and without using the divide-and-conquer approach were provided in Extended Data Figure 12. Also, the computed confidence scores were presented in the form of a graph in Extended Data Figure 10 and correlation coefficients were given in the main text. Regarding molecular interactions, we provide computed results of binding energies of molecules to proteins and discuss the influence of cut-offs on these results.

Furthermore, we want to point out that the applications in this work for which the framework was used and tailored for (confidence score calculation and spectral investigations) were tested before. For this we refer to the sections “Spectral calculation parameters” and “Atomic energy processing” in the methods part and to Extended Data Figures 11, 12 and 13. Additionally, the spectral calculations were compared to experimental results (references 42, 43, 44) in the section “Quantum optical spectra of biomolecules” and in Figure 4.

“1. The authors have performed no structural optimisation or molecular dynamics in the present paper. This is an important issue because I am concerned that their truncated ion-ion interaction will lead to wrong forces, and hence wrong structures or dynamics. Similarly, the lack of dispersion at even an empirical level will have an effect. Ideally this would be tested, but it certainly must be highlighted.”

We performed calculations which show that the truncated ion-ion interaction and the employed Coulomb cut-off scheme do not lead to wrong forces, structures or dynamics. For that, we present results for all of these topics with and without using the Coulomb cut-off scheme which show only very small deviations from each other. While we consider molecular dynamics outside the scope of our work, we agree that these results will be an additional validation of the formalism. Results from calculations without dispersion are discussed in point #4. In the following, we provide detailed information about calculations regarding structural optimization and molecular dynamics.

As a test system to study structural optimization and molecular dynamics, we used a DNA tetramer (sequence ACGT). This system also fulfils the conditions in point #2 of a charged and polarized structure since DNA is negatively charged due to the phosphate groups in its backbone and it has a higher polarizability than proteins. Furthermore, the system is large enough to show effects of the truncation of the ion-ion interaction and still has a size for which it is possible to perform MD simulations without cut-offs at a reasonable computational effort.

Here, we calculated the forces for this structure with and without using the Coulomb cut-off. The parametrization of the Coulomb cut-off is the same for the electron repulsion integrals (ERIs) and for the ion-ion interactions. In Figure 14 (see below) it is shown that the forces from a calculation without cut-off are reproduced well in a calculation with the Coulomb cut-off. The figure numbers are taken from the manuscript and are therefore not always ordered in this reply letter.

Figure 14: Forces for DNA (ACGT) with and without Coulomb cut-off. For a DNA molecule (ACGT) the atomic forces are displayed without (x-axis) and with a Coulomb cut-off (y-axis) used. Two different Coulomb cut-offs configurations are tested for the lower and upper Coulomb cut-off used in this work. The mean absolute errors are 4.2 % and 2.5 %, respectively.

In addition to this, a structural optimization was performed for the molecule. For this the Broyden–Fletcher–Goldfarb–Shanno (BFGS) algorithm was used – a quasi-Newton method which uses an approximation of the Hessian and which is commonly used for structural optimization or for nonlinear optimization problems in general. Compared to gradient descent methods which do not precondition the gradient vector, the BFGS algorithm usually shows much faster convergence. However, a convergence to zero or values very close to zero is usually not achieved for larger molecules since this would also require to perform a structural optimization at a larger scale. Therefore, the root-mean-square (RMS) of the forces is often improved by one or two orders of magnitude until the algorithm plateaus. This is also discussed in reference 16.

The development of the RMS of the forces during a BFGS optimization of the ACGT molecule is shown in Figure 16 for two calculations without and with Coulomb cut-off (cut-off values at 3.5 and 10 Angstrom), respectively. The calculation was run over 30 optimization steps. In each iteration, the optimal descent prefactor was determined using a line search utilizing the golden section search algorithm which avoids re-evaluations as much as possible. During the line search, only the energy is needed and not the forces and therefore the programme presented in this work was run in energy-only mode again to save computation time. In Figure 16 we observe convergence for the calculations with and without cut-off. Both converge in a very similar way and show oscillatory convergence at the end. The minimum RMS is achieved at step 28 with a value of around 0.0009 for both calculations.

Figure 16: Force RMS development under geometry optimization with and without cut-off for DNA (ACGT). A geometry optimization has been performed for a DNA tetramer (sequence ACGT) which uses the BFGS method with a corresponding golden section line search and ran over 30 optimization steps. The convergence to an equilibrium structure is evaluated by computing the root-mean-square of the computed forces at each optimization step.

In Figure 17 the atomic coordinates from before and after the structure optimization are shown as an overlay. Additionally, at the bottom, the final coordinates from the structural optimization with Coulomb cut-off are overlaid with the final coordinates from the optimization without Coulomb cut-off. Here, we observe that the structural optimization does have an influence on the geometry of the system (see top panel of Figure 17), as it is expected. Furthermore, the final geometry is not significantly influenced by the Coulomb cut-off as both calculations lead to a very similar result. The root-mean-square displacement between the two calculations is computed as 0.047 Angstrom. From these calculations we can conclude that the Coulomb cut-off only has a small influence on structural optimization and certainly does not lead to unstable structures, extremely large forces or other physical or numerical instabilities.

It should also be mentioned that forces that deviate from the forces without cut-offs (both density and Coulomb cut-off) still are the derivative of a (now slightly changed) energy landscape. Nonetheless, the sum of the forces still remains zero as they are the exact derivative of the energy. This has been tested by ensuring that the forces obtained from the programme can be reproduced numerically as the limit case of the finite-differences method. Testing has been done for the forces as a sum of derivatives as well as for the derivatives of the integrals itself (overlap, kinetic, nuclei and electron repulsion integrals).

Figure 17: Overlays of atomic coordinates from structural optimization. Top: The atomic coordinates from a DNA molecule (ACGT) from before the structural optimization are overlaid with the coordinates after the structural optimization with no Coulomb cut-off used. Bottom: The coordinates after structural optimization with no Coulomb cut-off used are overlaid with the coordinates after structural optimization with a Coulomb cut-off used with Coulomb cut-offs of 3.5 and 10 Angstrom for the lower and upper Coulomb cut-off, respectively.

We continue the testing by investigating the stability of molecular dynamics simulations with and without the Coulomb cut-off. To do so, we performed an MD simulation of the same system that was used above for the investigation of the structural optimization process. The simulation runs over 200 time steps with a step interval of 0.2 fs.

It has to be mentioned that a slight drift in the total energy can be seen with a value of 6.0×10^{-7} Ha per atom and per femtosecond, this is caused by some of the starting forces being very large and is not an error which occurs due to cut-offs as it happens in the calculation without cut-offs as well. The MD simulation was done with large forces to test the performance of the cut-offs for more extreme situations beyond the equilibrium – for this reason, we also chose to work with the rather drastic cut-off scheme 3.5/10. However, to ensure that drift of the total energy is indeed caused only by the integrator, a simulation was done with a smaller time step for a smaller system (DNA with just one base pair, cytosine) with the Coulomb cut-off used. In this case it could be observed that the changes of the total energy can be reduced by decreasing the time step of the molecular dynamics simulation. To be precise, reducing the time step ten-fold leads to an energy conservation error of 8.0×10^{-8} Ha per atom and per femtosecond. For an MD simulation with 10 steps of geometry optimization prior to its start, a time step of 0.5 fs can be used with an energy conservation error of 2.9×10^{-9} Ha/atom/fs.

Figure 18: Energy development during an MD simulation of DNA (ACGT) with and without cut-offs. A molecular dynamics simulation was performed for the DNA tetramer with the sequence ACGT. The development of the total, potential and kinetic energy is shown for the same simulation without (left) and with (right) a Coulomb cut-off used. For the simulation that uses a Coulomb cut-off, the values of the lower and upper Coulomb cut-off are 3.5 and 10 Angstrom respectively.

Here, we observe very little difference between the kinetics in the two MD simulations with and without cut-off used. Of course, this is not a full test, therefore we also investigate the actual trajectories. However, from Figure 18 it can be seen that the low Coulomb cut-off does not lead to numerical instabilities or to larger forces than in the simulation with cut-off.

Next, we consider the trajectories obtained in the two simulations. We computed the RMSD between the coordinates of the two simulations at each time step. This is shown in Figure 19

below as the green line. Additionally, the mean travelled distance of the atoms in the simulation was computed for both simulations is also displayed in Figure 19.

Figure 19: Trajectory comparisons of MD simulation of DNA (ACGT) with and without cut-offs. For a molecular dynamics simulation of the mentioned DNA tetramer, the mean travelled distance is plotted for the same simulation with and without a Coulomb cut-off used. Additionally, the root-mean-square displacement between the coordinates of these two simulations is plotted for each time step. For the simulation that uses a Coulomb cut-off, the values of the lower and upper Coulomb cut-off are 3.5 and 10 Angstrom, respectively.

Here, we observe a difference in the trajectories of the two MD simulations but it is small compared to the total movement of the atoms. The maximum RMSD is 0.091 Angstrom. In Figure 20 the atomic coordinates from the start and the end of the MD simulation with no cut-off are overlayed and a second overlay is shown where the final MD coordinates of the simulation with no cut-off are compared with the final coordinates of the MD simulation with the Coulomb cut-off.

Figure 20: Overlays of atomic coordinates from an MD simulation of DNA (ACGT). In a similar fashion to the results depicted in Figure 17, the coordinates are shown before and after an MD simulation. Top: Atomic coordinates from before and after an MD simulation of 200 time steps are overlaid. No Coulomb cut-offs were used in this calculation. Bottom: Overlay of the atomic coordinates after two MD simulations with identical starting conditions, one

with Coulomb no cut-off and another one with Coulomb cut-off (3.5 and 10 Angstrom for the lower and upper cut-off).

Here, we observe again that the changes due to the cut-off are small compared to the total movement in the MD simulation. Note that here a quite aggressive cut-off configuration (3.5 and 10 Angstrom for the lower and upper cut-off) is used which is not recommended due the very small value of 3.5 Angstrom for the lower cut-off. In the following paragraph we discuss the reason for the large deviation between the lower and upper cut-off and why a higher cut-off value is recommended for force calculations.

The choice of the lower and upper cut-off plays an important role specifically for force calculations. For the calculations that only compute the energy of the system, the difference between the two cut-offs only has the purpose to smoothen the transition to zero. However, if the energy gradient is computed, a fast transition from the actual value of the Coulomb interaction to zero can lead to very large derivatives. This can lead to the derivative between the two cut-off values being larger than the derivative for distances below the lower cut-off value. Such a behaviour has to be avoided to maintain a smooth transition to zero for the forces as well. This is achieved by increasing the distance between the values of the two Coulomb cut-offs, ideally the upper Coulomb cut-off should be at least 2.5 times larger than the lower cut-off to ensure a smooth transition to zero for the forces. For this reason, a cut-off configuration of 8 / 20 is more recommended which was tested at the beginning of this section for the force comparisons.

Finally, regarding the effects of the neglect of correlation we want to forward to the results for point #4 where binding energies were computed which give reasonable results even without correlation. Of course, Coulomb correlation still plays an important role in biophysical systems. Therefore, including empirical correlation might be a target for future work, for example using the HF-3c formalism by Sure & Grimme (J. Comput. Chem., 34:1672–1685, 2013) which provides good corrections while still using a small basis set. For this work however, we consider these corrections outside of the scope of the work, especially since the presented applications lead to matching results with other data without correlation included. Missing correlation is of course still a limitation and therefore this is discussed in the paragraph “Strengths and limitations of the method” in the main text.

“2. The lack of communication between subsystems in the divide and conquer method (and the lack of total electron number conservation that the authors allude to) strikes me as a significant issue, particularly for structures which are charged or polarised, and might well affect optimisation or dynamics. Some form of testing would be helpful.”

It has been tested that the lack of communication between the subsystems does not influence the results in calculations using the divide-and-conquer method significantly. In the first submission to Comms Chem these test results are displayed in Figure 11. Additionally, we

performed calculations of forces with and without divide-and-conquer used which are in good agreement with each other. These results are discussed in detail on the following pages.

For this point we performed the same test as with point #1 where the forces are computed with and without the approximation used. Here we study a larger system in the form of the DNA 12-mer ACGTACGTACGT which will be split into three separate sections during the divide-and-conquer formalism. We compute the forces using the Coulomb cut-off configuration (3.5/10) discussed in point #1.

Two different configurations are tested, one with a buffer zone of 8 and one with 10 Angstrom. Neighbouring atoms to the buffer zone and hydrogen atoms for bond termination have been added as described in the respective section in the main text.

Figure 15: Forces for DNA (ACGT) with and without divide-and-conquer. For a DNA molecule (ACGT) the atomic forces are displayed without (x-axis) and with the divide-and-conquer method (y-axis) used. Two different buffer zone configurations are tested (8 and 10 Angstrom). The mean absolute errors are 0.25 % and 0.047 %, respectively.

From Figure 15 we can see that the errors of the forces are quite low, even for the smaller buffer zone at 8 Angstrom. One explanation for this is that the Coulomb cut-off already truncated a lot of the long-range interaction but nonetheless this test gives the information that no significant additional errors occur on top of those caused by the Coulomb cut-off.

Furthermore, we see that the distribution of the electron density in the core region is not significantly influenced by the parts not included in the buffer zone because otherwise stronger deviations in the forces would be expected. Another explanation for the low error is the adding of neighbouring atoms to avoid breaking bonds and the replacement of heavier atoms with hydrogen atoms at the section borders which help to construct physically sensible boundary conditions, potentially reducing errors further.

Furthermore, we want to refer to another test which was performed for the protein Evasin which was investigated as one of the three structures for which the quantum-mechanical confidence scores were computed. Here, the atomic energies, which are used as the discussed quantum-mechanical confidence score, are computed with and without using the divide-and-conquer formalism. This was done as a test tailored to investigate the effects of divide-and-conquer on the confidence score calculations as one of the main applications presented in this manuscript. The results in Figure 11 below indeed demonstrate that there are only very minimal difference between the results obtained with and without using divide-and-conquer for the quantity studied in this specific application.

Figure 11: Atomic energies with and without divide-and-conquer. Computed atomic energies are displayed for Evasin P1126, one of the AlphaFold predicted protein structures investigated in this work. The system was studied without water, otherwise a comparing calculation without divide-and-conquer would be computationally too demanding. The energies without using the divide-and-conquer method are plotted in blue and with a larger linewidth for a better visualization (to prevent too much overlap between the two lines) than the energies with divide-and-conquer in orange.

“3. I am concerned that Figure 5, one of their key results, is not a sensitive test of this method (as indicated by Referee 3 in their report). No colour map is given, and it is very hard to get any quantitative measure of deviation by comparing visually like this.”

This is a good point; we agree that the colours alone do not offer a detailed representation of the results. Answering to a point raised by reviewer 2 during the review round in Nat Comm, we already added an extended data figure in the first submission to Comms Chem, which presents the results in the form of a graph. This figure can be seen below.

Figure 12: AlphaFold pLDDT score in comparison with atomic energies. The pLDDT scores by AlphaFold (blue) and the predicted pLDDT scores computed with the described method using atomic energies (orange) are compared for the three predicted protein structures investigated in this work.

Additionally, reviewer 2 suggested calculating the Pearson correlation coefficient which we added to the main text – for the three investigated proteins the Pearson correlation coefficients between the predicted pLDDT score and AlphaFold’s pLDDT score are computed as 0.953, 0.914 and 0.946, respectively.

Since the report from reviewer 3 was also mentioned: They pointed out in their report that “The correlation with Alphafold pLDDT scores depends simply on local packing effects.” The role of packing effects is indeed an important point and was therefore discussed already in the first submission in lines 251 ff. and in extended data Figure 10. From this separate calculation of packing densities, it can be concluded that this comment is not correct. While there of course is a correlation with packing densities, the correlation between the two confidence scores is much stronger. This is not completely unexpected as protein folding is a problem of energy minimization and not of maximizing packing densities.

Figure 10: Atomic energies versus neighbouring atom count. For the predicted protein structure PDB AF_AFA0A023IKK2F1⁴⁷ (a), the structure evaluation method using atomic energies from Hartree–Fock is compared with an approach that uses the number of neighbouring atoms as a metric. Higher local atom density is typically associated with more accurate predictions, resulting in a loose correlation between the neighbouring atom count (b) and AlphaFold’s pLDDT scores (d). However, the evaluation of protein structures with Hartree–Fock atomic energies (c) provides a more accurate assessment. For example, the alpha helix at the bottom left of the structure is correctly assigned high confidence by the quantum approach but not by the neighbour count method. Similarly, the linker between the two top-left domains is identified as stable by the Hartree–Fock energies but misclassified as a weak region by the atom density method.

To summarize, the correlation coefficients and the visualization of the two different confidence scores as a graph should provide a quantitative test of the presented method.

“4. Any interactions between molecules (as suggested in the first sentence of the introduction) will be strongly affected by the types of problem I suggest above. Since a major application of this method seems to be this kind of question, some characterisation of the limitations is needed.”

Interactions between molecules can still be studied, even if minimal-basis Hartree-Fock is used. For that, we performed two calculations which show agreement with experimental data. This test is described in the following section, including a discussion of the influence of cut-offs.

For this point we consider binding energies between two molecules and two proteins as two test systems. To be precise, the two systems studied are the vitamin biotin (vitamin H) bound to the protein avidin and the drug FK506 bound to the protein FKBP. Both of these systems have high binding energies of -20.4 and -12.8 kcal/mol, respectively (see references 94 and 95). Since the computation of binding energies can be considered an area where high accuracies are necessary, it serves as a test if minimal-basis Hartree-Fock can still provide somewhat reliable results. This is also related to point #1 where the missing correlation in Hartree-Fock calculations is discussed and therefore investigates if HF as an uncorrelated method can still be useful in studying biophysical systems.

For this test we perform a single-point energy calculation using Hartree-Fock and again try out different cut-off configurations. Since the whole protein with ligand is too large to study without cut-offs, we focus on the area around the ligand. Therefore, the system used in the calculations consists of the ligand and the environment within 4 Angstrom around the ligand. Additionally, bond cutting is accounted for by adding further atoms and replacing some atoms with hydrogen atoms to terminate bonds. The exact procedure is described in the corresponding section in the main text and the resulting systems as well as the complete protein-ligand complex are shown in Figure 21 below.

Figure 21: Systems studied for binding energy tests. Two systems were used to compute the binding energy between a protein and a ligand. The first system (top row) is biotin bound to avidin (PDB 1AVD) and the second system (bottom row) is the drug FK506 bound to the protein FKBP (PDB 1FKJ). On the left side the protein is shown as a ribbon diagramme and the ligand is depicted in black. The ligand is studied in an environment of 4 Angstrom with neighbouring atoms added according to the procedure described in the corresponding section. The resulting structures for the two systems are shown on the right side.

For the two systems discussed above, the binding energy is computed as the energy difference between the energy of the protein-ligand complex and the sum of energies of the protein and the ligand (with “protein” we here refer to the part of the protein treated in the calculation, as described above). In Table 1 (see below) the resulting energies are listed for different combinations of density and Coulomb cut-offs.

System	Density cut-off	Coulomb cut-off [Angstrom]	Computed binding energy [kcal/mol]	Experimental binding energy [kcal/mol]
Biotin/Avidin	1.0e-6	None	-18.97	-20.4
Biotin/Avidin	1.0e-7	None	-19.23	-20.4
Biotin/Avidin	1.0e-8	None	-19.23	-20.4
Biotin/Avidin	1.0e-5	None	-28.07	-20.4
Biotin/Avidin	1.0e-6	3.5 / 10	-8.86	-20.4
Biotin/Avidin	1.0e-6	8 / 10	-41.86	-20.4
Biotin/Avidin	1.0e-6	15 / 20	-18.62	-20.4
FK506/FKBP	1.0e-6	None	-15.11	-12.8
FK506/FKBP	1.0e-7	None	-15.06	-12.8
FK506/FKBP	1.0e-8	None	-15.07	-12.8
FK506/FKBP	1.0e-5	None	-15.06	-12.8
FK506/FKBP	1.0e-6	3.5 / 10	0.96	-12.8
FK506/FKBP	1.0e-6	8 / 10	18.89	-12.8
FK506/FKBP	1.0e-6	15 / 20	-17.54	-12.8

Table 1: Binding energies for two test systems for different cut-off configurations. For biotin bound to avidin (PDB 1AVD) and FK506 bound to FKBP (PDB 1FKJ) the binding energy is computed for different configurations.

Here, we see limitations due to the use of cut-offs. Especially for the Coulomb cut-off configurations 3.5 / 10 and 8 / 10, the binding energy shows strong deviations from the results with no Coulomb cut-off – in some cases no binding is predicted. A reduced density cut-off of 1.0e-5 also show a deviation for the biotin/avidin complex. A higher Coulomb cut-off configuration (15 / 20) shows less strong deviation from the binding energy without a Coulomb cut-off. Lowering the density cut-off to values of 1.0e-7 or 1.0e-8 has little influence on the computed binding energy. However, for the calculations with the highest accuracy, the experimental binding energies can be reproduced with errors of 2.3 and 1.2 kcal/mol.

Regarding the characterisation of limitations discussed in this section, we also want to refer to the testing of the real-time time-dependent Hartree-Fock calculations which were included in the second submission. These also provide a discussion about which thresholds should be used for the density and the Coulomb cut-off in absorption spectra calculation. The relevant figure (Figure 13) is added below.

Figure 13: Absorption spectra for beta-carotene with different parametrizations. The computed absorption spectrum of the molecule beta-carotene ($C_{40}H_{56}$) using RTHF. Four different configurations are shown: no density cut-off and no Coulomb cut-off (blue), density cut-off 10^{-6} and Coulomb cut-off 10 \AA (orange), density cut-off 10^{-3} and Coulomb cut-off 10 \AA (green), density cut-off 10^{-6} and Coulomb cut-off 6 \AA (red). The second parametrization is used for the UV/Vis spectra in this work and shows little deviation from the calculation with no cut-offs.

Therefore, we can show that even plain Hartree-Fock offers possibilities to describe the interactions of molecules. The limitations of the framework with respect to the cut-offs have been described for calculations which require a higher accuracy. For the presented applications in this work, the used parametrization is tested to ensure performance at a sufficient accuracy for the scope of this work.

In summary, we think that adding further tests of the method was useful to assess its validity and applicability. Of course, we performed extensive testing prior to submitting the manuscript to be able to support the reliability of the presented methods for the presented applications. Additionally, we do not want to make general claims for the accuracy of the methods presented in this work beyond the use cases discussed in this manuscript and hope that is communicated correctly in the revised manuscript – if not, we are of course willing to change the respective sections. In general, the work should be seen as a proof of concept for the presented large-scale calculations with selected applications for which the formalism has been tested. The additional information about the testing process which is now included in the

third version of the manuscript should help to prove the reliability of the framework. For the specific applications, their needed accuracy and in general the scope of this work as described above we therefore consider the presented framework as reliable and well-tested. Figures 14-21 from this reply letter as well as Table 1 have been added into the Extended Data section at the end of the manuscript and the additional content has been described in an additional section in the Methods part of the manuscript.

Reply letter for the final submission

Manuscript: A Quantum-Mechanical Framework for Million-Atom Scale Biological Systems
Luc Wieners & Martin E. Garcia

We thank the reviewer for their comments in this review round and throughout the entire review process as well as for recommending the paper for publication.

In the following, we provide replies to the comments and the changes made in the manuscript.

“I would commend the authors for their extensive work to check the accuracy of their method. I was a little surprised at how insensitive the simple molecular relaxations and dynamics were to the cutoffs, but this is useful data.

However, the results in Table 1 in the rebuttal letter and the extended data highlight my key concern with the implementation of divide and conquer. I disagree with the authors' interpretation: I do not think that they can claim any kind of convergence here with the Coulomb cutoff - the oscillations in binding energy are still significant. I think that there should be some discussion of this in the text.”

A discussion of the divide-and-conquer method based on the results in Table 1 is not possible since the respective calculations did not use divide-and-conquer.

However, we included a general statement about the limitations of using the divide-and-conquer approach and the Coulomb cut-off in the “Strengths and limitations” section at the beginning of the main text:

“It should be mentioned that the use of the divide-and-conquer approach is focused on the applications in this work, for other areas – especially where higher accuracy is needed – the truncation of long-range interactions could potentially compromise the results. For systems with hundreds to a few thousand atoms, where no divide-and-conquer is needed, the low Coulomb cut-off leads to similar problems. In cases where smaller systems are studied at a higher accuracy, it is therefore necessary to work without divide-and-conquer and no (or a much larger) Coulomb cut-off. Test results can be found in the section “Further testing with molecular dynamics” in Methods.”

We agree that Table 1 shows the limitations of the implementation. These were already discussed in the previous submission in the following way:

“Here, we see limitations due to the use of cut-offs. Especially for the Coulomb cut-off configurations 3.5 / 10 and 8 / 10, the binding energy shows strong deviations from the results with no Coulomb cut-off – in some cases no binding is predicted. A reduced density cut-off of 1.0e-5 also show a deviation for the biotin/avidin complex.”

Table 1 was rather intended as “a test if minimal-basis Hartree-Fock can still provide somewhat reliable results”. In the revised manuscript, we clarified that no divide-and-conquer was used and we give an interpretation of the results:

“These results show that using low Coulomb cut-offs makes it impossible to use the formalism for applications which need a higher accuracy. In these cases, it is necessary to use no (or a much larger) Coulomb cut-off. The density cut-off at 10^{-6} appears to be a reasonable choice which should not compromise the accuracy significantly. Consequently, testing is required if a Coulomb cut-off or a higher density cut-off is used for applications beyond those studied in this work to ensure that the accuracy is sufficient for the investigated problems. If no Coulomb cut-off is used and a density cut-off smaller than 10^{-6} is employed, we observe that we can reproduce the experimental binding energies for the two studied systems within a reasonable error.”

“Moreover, I think that some of the claims made in the abstract and introduction need to be removed or toned down; in particular in lines 29-31 (“This approach opens new avenues in quantum physics, structural biology, spectroscopy, bioinformatics, medicine, and materials science.”) and lines 50-53 (“This advance bridges quantum mechanics and biology at a very high computational speed, enabling large-scale, first-principles simulations with broad applications in quantum biology, structural biology, medicine, materials science, and many-body physics.”) I do not think that the present results justify such wide-ranging claims: in particular it's not clear that molecule-molecule interactions are reliable with the Coulomb cutoff, and there is no evidence presented on the efficacy of the method in materials science or related areas.”

Both statements (lines 29-31 and lines 50-53) were removed entirely.